**REPORT**

# Mechanical control of osteoclast fusion by membrane-cortex attachment and BAR proteins

Yumeng Wan[1]* , Yuri L. Nemoto[1,2]* , Tsukasa Oikawa[3] , Kazunori Takano[4] , Takahiro K. Fujiwara[5] , Kazuya Tsujita[1,2] , and Toshiki Itoh[1,2]

**Osteoclasts are multinucleated giant cells that are formed by the fusion of precursor cells. Cell–cell fusion is mediated by membrane protrusion driven by actin reorganization, but the role of membrane mechanics in this process is unknown. Utilizing live-cell imaging, optical tweezers, manipulation of membrane-to-cortex attachment (MCA), and genetic interference, we show that a decrease in plasma membrane (PM) tension is a mechanical prerequisite for osteoclast fusion. Upon RANKL-induced differentiation, ezrin expression in fusion progenitor cells is reduced, resulting in a decrease in MCA-dependent PM tension. A forced elevation of PM tension by reinforcing the MCA conversely suppresses cell–cell fusion. Mechanistically, reduced PM tension leads to membrane protrusive invadosome formation driven by membrane curvature-inducing/sensing BAR proteins, thereby promoting cell–cell fusion. These findings provide insights into the mechanism of cell–cell fusion under the control of membrane mechanics.**

## Introduction

Osteoclasts are multinucleated giant cells that are formed by the fusion of precursor cells of the monocyte/macrophage lineage (Boyle et al., 2003; Teitelbaum and Ross, 2003). Upon binding to receptor activator of nuclear factor κB ligand (RANKL) on the cell surface of osteoblasts and osteocytes, osteoclast precursor cells become immature mononuclear osteoclasts (Yasuda et al., 1998). As they mature, multinucleated osteoclasts are formed by cell–cell fusion through the interaction of plasma membrane (PM) proteins, including dendritic cell–specific transmembrane protein (DC-STAMP) (Kukita et al., 2004; Yagi et al., 2005). Recent in vivo imaging studies have revealed that osteoclasts are not a static entity but rather exist in a state of dynamic equilibrium (McDonald et al., 2021). This equilibrium is maintained through the division of multinuclear osteoclasts into mononuclear cells designated as "osteomorphs," which subsequently fuse to re-establish the original multinucleated state. Therefore, the physical status of intact osteoclasts is highly dynamic and contingent upon their life cycle and physiological activities. This necessitates a detailed understanding of the cellular mechanics underlying cell–cell fusion.

The physical properties of cells are dependent on the formation and reorganization of the actin cytoskeleton beneath the PM. It has been proposed that actin-based invasive membrane protrusions at the cell–cell fusion interface play a crucial role in a wide range of cell fusion processes, including osteoclast and muscle fusion (Sens et al., 2010; Oikawa et al., 2012; Shilagardi et al., 2013; Søe et al., 2015; Wang et al., 2015; Faust et al., 2019; Kim and Chen, 2019). Moreover, Tks5, a central adaptor molecule that promotes WASP–Arp2/3 complex-dependent actin polymerization in the formation of invadosomes (a collective term encompassing invadopodia and podosomes), is indispensable for osteoclast fusion (Oikawa et al., 2012). In order for Arp2/3-dependent branched actin to develop protrusive membrane structures such as invadosomes, it is necessary to overcome the mechanical rigidity of the PM. Normally, the PM is subject to physical forces such as tension, which is primarily mediated by the adhesion between the PM and the underlying actin cortex, referred to as membrane-to-cortex attachment (MCA) (Sheetz, 2001; Gauthier et al., 2012; Diz-Muñoz et al., 2018). The ezrin-radixin-moesin (ERM) proteins are the major players in the establishment of the MCA. Recent studies have revealed that MCA-dependent PM tension mechanically controls cell membrane dynamics, including membrane protrusions (Sitarska and Diz-Muñoz, 2020; Itoh and Tsujita, 2023). These observations suggest the existence of a regulatory mechanism based on membrane mechanics that governs alterations in the PM morphology, including invadosome formation, during cell–cell fusion.

In this study, we investigate the relationship between MCA and osteoclast fusion by using cell biological and biophysical

[1]Division of Membrane Biology, Department of Biochemistry and Molecular Biology, Kobe University Graduate School of Medicine, Kobe, Japan; [2]Biosignal Research Center, Kobe University, Kobe, Japan; [3]Department of Molecular Biology, Graduate School of Medicine, Hokkaido University, Sapporo, Japan; [4]Department of Biology, Graduate School of Science, Chiba University, Chiba, Japan; [5]Institute for Integrated Cell-Material Sciences (WPI-iCeMS), Kyoto University, Kyoto, Japan.

*Y. Wan and Y.L. Nemoto contributed equally to this paper. Correspondence to Toshiki Itoh: titoh@people.kobe-u.ac.jp; Kazuya Tsujita: tsujita@people.kobe-u.ac.jp.

approaches. Our findings indicate that a reduction in MCA is a prerequisite for both of the two fusing cells during osteoclast fusion. Additionally, two groups of BAR proteins were identified as essential factors for cell–cell fusion in response to MCA decrease. We propose that MCA serves as an inhibitory regulator of invadosome formation, thereby underscoring its significance as a physical regulator of osteoclast fusion.

## Results and discussion

### RAW 264.7 cells transform to a well-spread morphology before RANKL-induced cell–cell fusion

To investigate the role of MCA in osteoclast fusion, we first monitored the behavior of the murine leukemia macrophage RAW 264.7, which had been induced to differentiate into osteoclasts by RANKL treatment (Hsu et al., 1999; Oikawa et al., 2012; Kim et al., 2025). A time-lapse observation demonstrated that cells began to fuse with each other at 48 h (fusion index was 4.25 ± 4.49%, median ± SD, Fig. S1, A and B) after the addition of RANKL (Fig. 1, A and B; and Video 1). Notably, even with RANKL treatment, not all of the cells were involved in the cell fusion event. Consequently, mononuclear cells were classified into three categories based on their behaviors: (1) without RANKL treatment (therefore non-fused); (2) RANKL-treated, non-fused, which were treated with RANKL but did not exhibit cell–cell fusion until the end of the observation period; and (3) RANKL-treated, fused, which underwent cell–cell fusion (Fig. 1 B). For these three categories, we focused on the morphological changes of cells before and after cell–cell fusion by measuring the size (area) and circularity of individual cells over time (Fig. 1, C and D). The size of RANKL-treated and fused cells (fused) was slightly larger than that of other cells (non-fused, either with or without RANKL treatment) (Fig. 1 E, time ≤ −0.5 h, see Materials and methods section for details), reflecting a significant enlargement due to cell–cell fusion. Of note, the circularity of cells that eventually demonstrated fusion was constantly higher than those in other categories a few hours before the fusion event occurred (Fig. 1 F). We found that the fused cells exhibited significantly larger and more circular morphology than not only the non-treated cells but also the RANKL-treated, non-fused cells (Fig. 1, G and H). F-actin staining and subsequent observation by confocal microscopy revealed that while non-treated cells have filopodia, RANKL-treated cells show F-actin–enriched invadosomes at the extended as well as flattened membrane structure (Fig. S1 C). These results indicate that mononuclear cells have transformed from a spindle-like, filopodia-rich morphology to a well-spread, invadosome-dominant morphology before cell–cell fusion.

### RANKL-induced osteoclast differentiation reduces MCA to initiate cell–cell fusion

The correlation between cell–cell fusion and flattened and invadosome-dominant morphology led us to hypothesize that a reduction in MCA may be a prerequisite for the process of cell–cell fusion. Therefore, we used optical tweezers to assess the membrane tether force of RANKL-treated mononuclear cells (Fig. 2 A). It is known that tether force is proportional to the MCA-dependent PM tension (Sheetz, 2001; Gauthier et al., 2012). We found that the tether force of RANKL-treated cells was significantly lower than that of control cells (Fig. 2 B, parental).

If a reduction in MCA is a prerequisite for cell–cell fusion, an increase in MCA should conversely inhibit fusion. To test this hypothesis, we generated MCA high cells using recently developed molecular tools, membrane-targeted active ezrin (MA-ezrin) and signaling-inert MCA (iMC) linker (Fig. S2 A). These tools have a lipidation motif of Lyn (Lyn$_{10}$) for anchoring to the PM and a constitutive active form of ezrin (ezrin T567E) or the F-actin–binding domain of utrophin (Fig. 2 C) (Bergert et al., 2021; Tsujita et al., 2021). Consequently, these molecular tools constantly connect the PM and the actin cortex, reinforcing MCA. We found that expression of either MA-ezrin or iMC-linker significantly increased PM tension compared with the parental cell line, even in the presence of RANKL (Fig. 2 B, MA and iMC). In contrast, ezrin-expressing cells exhibited little effect on tether force (Fig. 2 B, Ezrin).

To confirm the effect of increased PM tension on cell–cell fusion, we examined the fusion efficiency of control (parental) and MA-ezrin– or iMC-linker–expressing cells after RANKL treatment. The results showed that cell–cell fusion was significantly suppressed by the expression of MA-ezrin or iMC-linker (Fig. 2, D and E). The expression of these tools did not inhibit the induction and the nuclear transport of nuclear factor of activated T cells cytoplasmic 1 (NFATc1), a master transcription factor of osteoclastogenesis (Fig. S2, B–E). In addition, all cell lines exhibited tartrate-resistant acid phosphatase (TRAP) activity, a specific marker of osteoclast differentiation, after RANKL treatment (Fig. S2 F), excluding the possibility that suppression of osteoclast fusion was due to the inhibition of their differentiation. These results demonstrate that increasing MCA-dependent PM tension is sufficient for the suppression of osteoclast fusion.

Previous studies have shown that asymmetric mechanical tension plays an important role in myoblast fusion and yeast gamete fusion (Kim et al., 2015; Muriel et al., 2021). To determine whether osteoclast fusion is facilitated by asymmetric PM tension between two fusing cells, we performed a co-culture experiment with MA-ezrin– or iMC-linker–expressing cell lines and their parental counterparts (Fig. 2 D). The fusion index of the mixed cells was significantly lower than that of the parental cells alone (Fig. 2 E), indicating that symmetric PM tension reduction in both cells, rather than PM tension asymmetry, is a critical factor in osteoclast fusion.

### Reduction of MCA upon RANKL-induced osteoclast differentiation is caused by decreased expression of ezrin

Next, we investigated whether ERM proteins are involved in osteoclast fusion. Western blot analysis revealed that among the ERM proteins in RAW 264.7 cells, only ezrin expression was decreased in response to RANKL treatment (Fig. S3, A–D). From an early point of cell–cell fusion (60 h) to its progression (84 h), the amount of ezrin was significantly reduced by RANKL treatment, as was the amount of phosphorylated ERM (p-ERM), the activated form of ERM proteins (Fig. 3, A–C). Consistently,

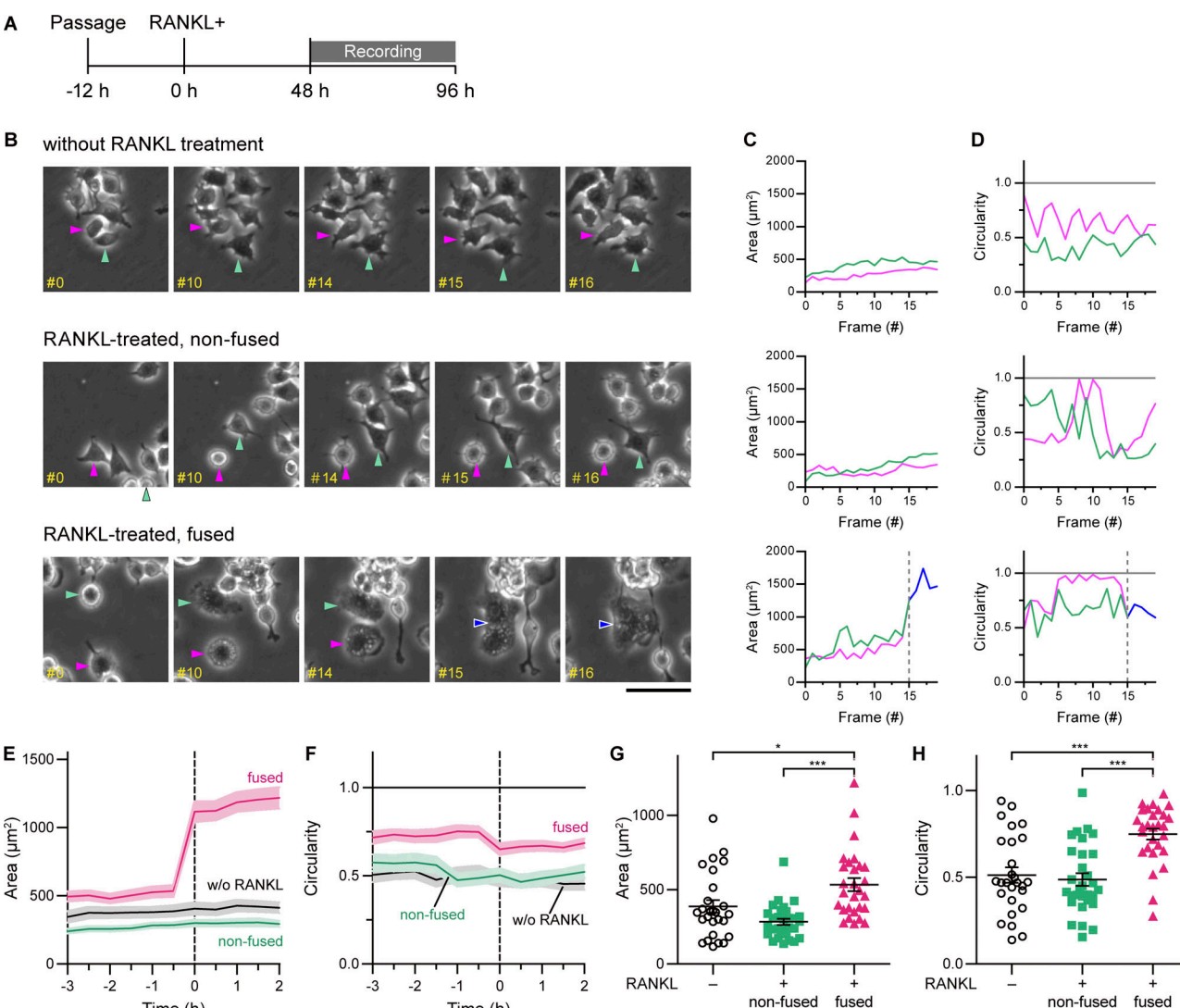

**Figure 1. RAW 264.7 cells transform to a well-spread morphology before RANKL-induced cell–cell fusion. (A)** Experimental timeline for time-lapse imaging. The frame rate is 30 min per frame. **(B)** Typical phase contrast image sequences of the cells without RANKL treatment (upper); RANKL-treated, non-fused (middle); and RANKL-treated, fused cells (lower). Arrowheads indicate the analyzed cells in the image. Magenta and green indicate mononuclear cells, and blue indicates a binuclear cell. The numbers in each image indicate frame numbers of sequential images taken every 30 min. Note that cell–cell fusion occurred at frame #15 of RANKL-treated, fused cells (lower). Scale bar: 50 μm. **(C and D)** The area size (C) and circularity (D) of each cell in B. The color of each line corresponds to each cell. The dotted line indicates the frame in which cell–cell fusion occurred. **(E and F)** Time-course changes of area size (E) and circularity (F) without RANKL treatment (black); RANKL-treated, non-fused (green); and fused cells (magenta). Time 0 was defined as the frame that cell–cell fusion occurred for RANKL-treated fused cells and an arbitrary time for non–RANKL-treated/RANKL-treated non-fused cells. Mean ± SEM, without RANKL: $n = 27$ cells, non-fused: $n = 30$ cells, and fused: $n = 28$ cells. **(G and H)** Scatter plots of area size (G) and circularity (H) at time −0.5 h, just before cell–cell fusion. Mean ± SEM, without RANKL: $n = 27$ cells, non-fused: $n = 30$ cells, and fused: $n = 28$ cells. P value obtained from one-way ANOVA with Tukey's test. *$P < 0.05$; ***$P < 0.001$.

the membranous p-ERM signal of RANKL-treated cells was reduced compared with that of control cells, (Fig. 3, D and E), suggesting that ezrin-mediated MCA plays a negative role in cell–cell fusion.

Inhibition of proteasome activity by MG132 did not affect the amount of ezrin, either before or after RANKL treatment (Fig. S3, E and F). Conversely, the level of *Ezr* (encoding ezrin) mRNA was reduced at 60 h after RANKL addition, whereas the level of *Nfatc1* mRNA was dramatically increased (Fig. S3, G and H). The gene expression of *Rdx* and *Msn* (encoding radixin and moesin, respectively) was not affected by RANKL treatment (Fig. S3, I

and J). These results suggest that the reduction of ezrin expression occurs through transcriptional control rather than through ubiquitin-mediated proteolysis.

B lymphocyte–induced maturation protein-1 (Blimp1), encoded by *Prdm1*, is induced by RANKL treatment via NFATc1 and plays a role as a transcriptional repressor downstream of NFATc1 during osteoclastogenesis (Nishikawa et al., 2010). In silico analysis revealed the presence of numerous Blimp1–binding sites within the promoter region of the mouse *Ezr* gene. Knockdown of Blimp1 tended to suppress the decrease in ezrin expression upon the addition of RANKL, although with no

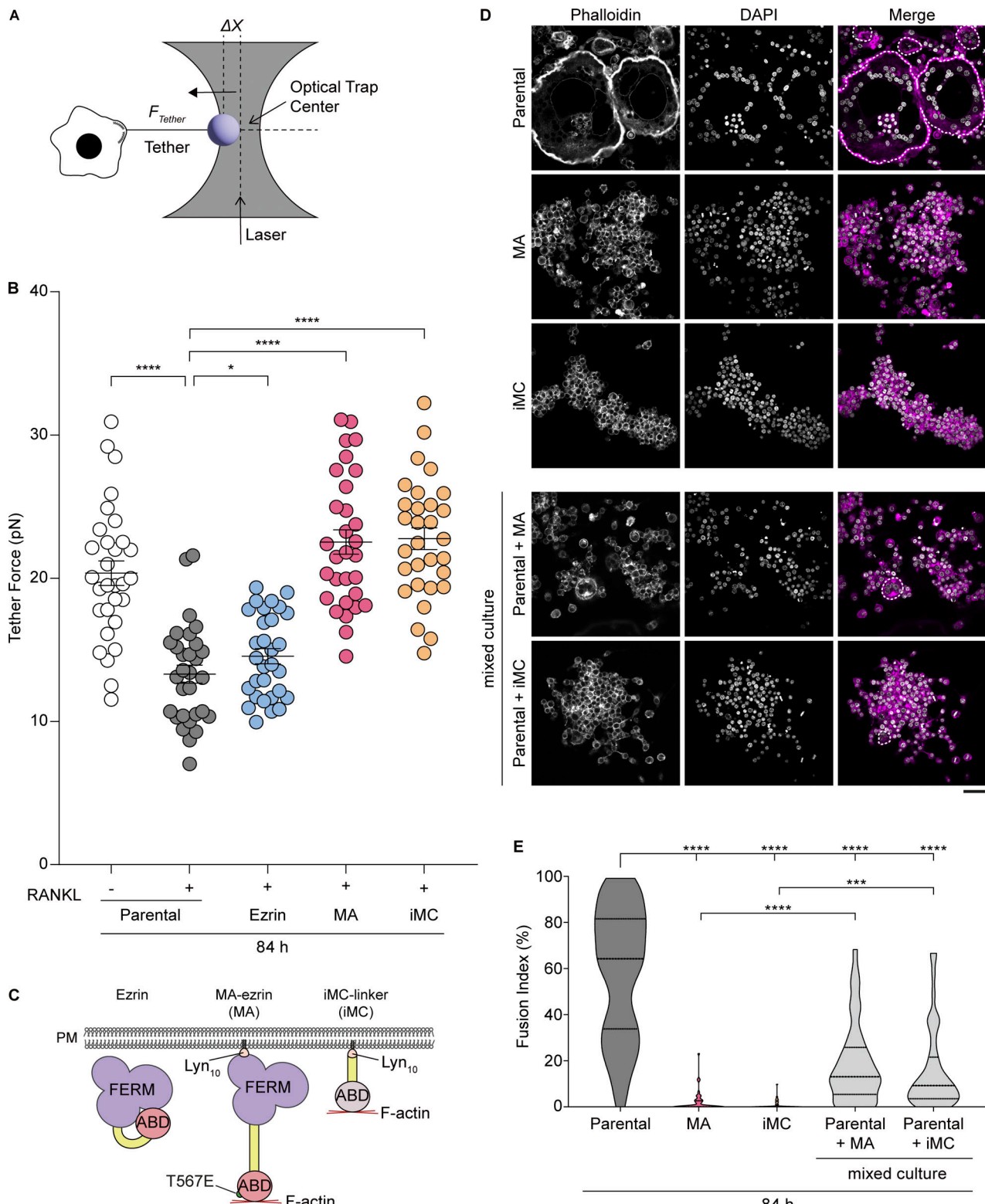

Figure 2. **RANKL-induced osteoclast differentiation reduces MCA to initiate cell–cell fusion. (A)** Schematic illustration of the measurement of tether force ($F_{Tether}$) using optical tweezers. $F_{Tether}$ can be calculated using Hooke's law, $F_{Tether} = k \times \Delta x$, where $k$ is the stiffness of the trap and $\Delta x$ is the displacement of the bead from the trap center. **(B)** Scatter plot comparing the tether force of mononuclear cells treated without or with RANKL for 84 h. Mean ± SEM, $n = 30$ cells for each condition. P value obtained from one-way ANOVA with Tukey's test. *P < 0.05; ****P < 0.0001. **(C)** Schematic of the molecules, wild-type ezrin (Ezrin), MA-ezrin (MA), and iMC-linker (iMC), overexpressed in RAW 264.7 cells. Lyn10, myristoylation sequence of Lyn; FERM, FERM domain of ezrin; ABD, the F-actin–binding domain. **(D)** Confocal images of cell lines in parental, MA-ezrin (MA), and iMC-linker (iMC) alone (upper) or in co-cultured (lower), after 84 h of RANKL treatment, stained with phalloidin (magenta) and DAPI (gray). Multinucleated cells are surrounded by dotted lines. Scale bar: 50 μm. **(E)** Quantification of fusion index in D. Dotted lines in violin plots show median and quantiles. $n = 45$ fields of view for each condition. P value obtained from one-way ANOVA with Tukey's test. ****P < 0.0001.

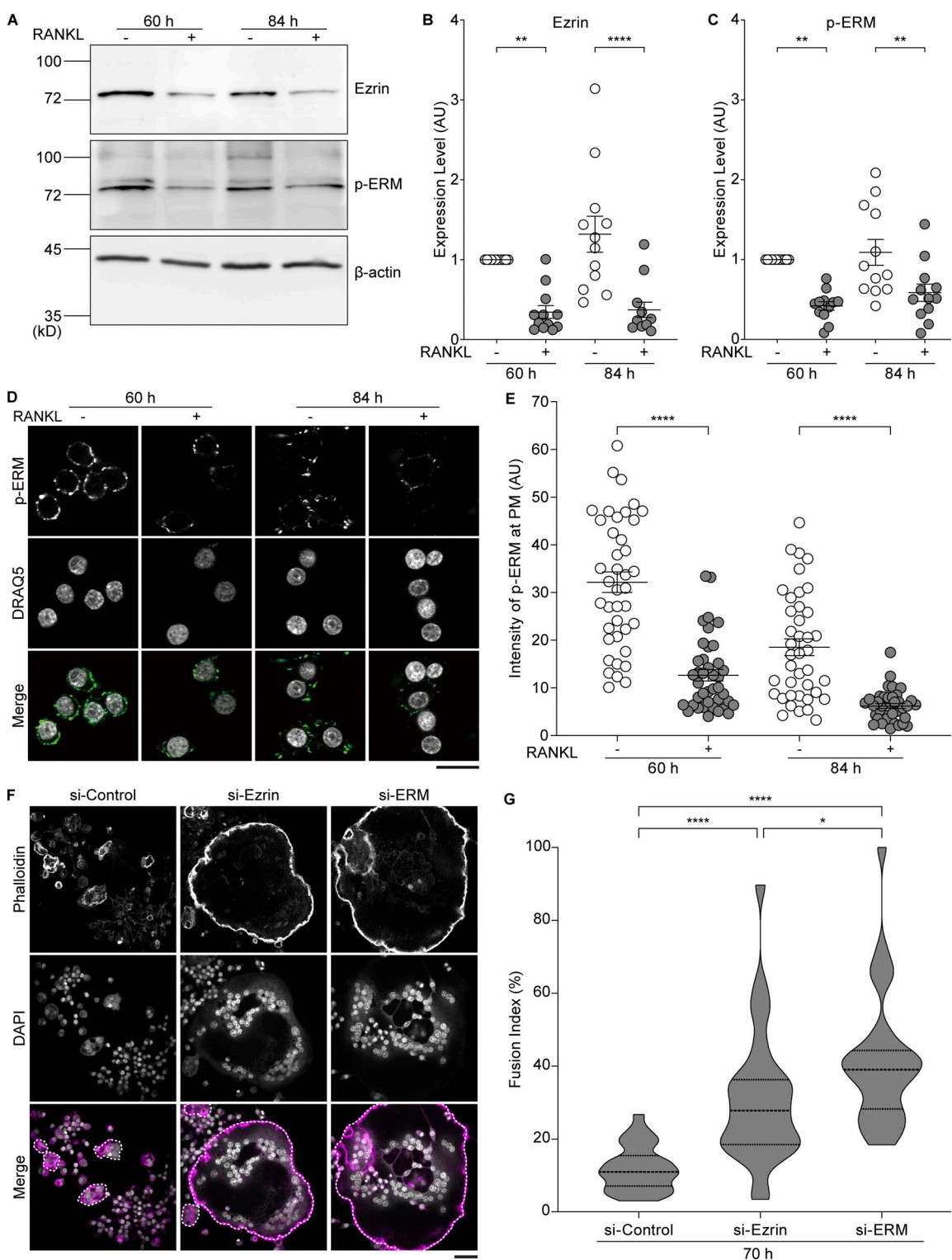

Figure 3. **Reduction of MCA by RANKL treatment is caused by decreased expression of ezrin. (A)** Western blot analysis of cell lysates from RAW 264.7 cells without or with RANKL treatments for the indicated times, using anti-ezrin, anti–p-ERM, and anti–β-actin antibodies. **(B and C)** Quantification of (A). Mean ± SEM, $n$ = 12 experiments. P value obtained from one-way ANOVA with Tukey's test. **P < 0.01; ****P < 0.0001. **(D)** Confocal images of RAW 264.7 cells without or with RANKL for 60 or 84 h, stained with anti–p-ERM antibody (green) and DRAQ5 (gray). Scale bar: 20 μm. **(E)** Quantification of fluorescence intensity of p-ERM beneath the PM determined by wheat germ agglutinin (WGA) signal. Mean ± SEM, $n$ = 40 cells for each condition. P value obtained from one-way ANOVA with Tukey's test. ****P < 0.0001. **(F)** Confocal images of siRNA knockdown cells treated with RANKL for 70 h, stained with phalloidin (magenta) and DAPI (gray). Multinucleated cells are surrounded by dotted lines. Scale bar: 50 μm. **(G)** Quantification of fusion index in F. Dotted lines in violin plots show median and quantiles. $n$ = 30 fields of view for each condition. P value obtained from one-way ANOVA with Tukey's test. *P < 0.05; ****P < 0.0001. Source data are available for this figure: SourceData F3.

statistical significance (Fig. S3, K and L; si-Control vs si-Prdm1 after the addition of RANKL), suggesting that additional transcription factors may also be involved in the suppression of ezrin expression.

If the reduction of ezrin expression plays an active role in cell–cell fusion, its depletion should facilitate fusion. To examine whether the fusion speed is accelerated, cells were fixed at an earlier time point. Indeed, the knockdown of ezrin in RANKL-treated cells resulted in a more than 2.5-fold increase in cell–cell fusion activity as early as 70 h after RANKL addition (Fig. 3, F and G, Fig. S3 M, and Video 2). Triple knockdown of the ERM proteins showed an ~30% increase in fusion activity when compared with the single *Ezr* silencing, suggesting that radixin and moesin also play a partial role in suppressing cell–cell fusion (Fig. 3, F and G; and Fig. S3 M). These findings suggest that the reduction of MCA is primarily dependent on the reduction of ezrin expression and accelerates cell–cell fusion.

### Knockdown of Baiap2, Baiap2l1, and the TOCA family proteins inhibits cell–cell fusion during RANKL-induced osteoclastogenesis

Next, we sought to identify effector proteins that facilitate cell–cell fusion downstream of MCA reduction accompanied by RANKL-induced osteoclast differentiation. BAR proteins are ideal candidates because their ability to sense and induce membrane curvature is known to be dependent on membrane tension (Tsujita et al., 2015; Itoh and Tsujita, 2023). We selected 14 BAR proteins expressed in RAW 264.7 cells and examined the effect of their depletion on cell–cell fusion (Fig. 4 A). Cell–cell fusion was reduced after knockdown of some BAR proteins, including members of the TOCA family (Fnbp1/FBP17, Fnbp1l/TOCA-1, and Trip10/CIP4), Baiap2/IRSp53, and Baiap2l1/IRTKS. This reduction was comparable with that observed for knockdown of DC-STAMP, a key regulator of cell–cell fusion during osteoclastogenesis (Fig. 4, A and B). Knockdown of Baiap2l1 and Fnbp1 (Fig. S3, N–P) resulted in a reduction of the fusion index, while TRAP activity remained unaffected (Fig. 4 C). This suggests that these five BAR proteins, namely the TOCA family (Fnbp1, Fnbp1l, and Trip10), Baiap2, and Baiap2l1, are required for cell–cell fusion without affecting osteoclast differentiation.

### Decreased MCA promotes invadosome formation for cell–cell fusion

Since invadosomes play a key role in cell–cell fusion (Oikawa et al., 2012), we next examined the relationship between invadosome formation and MCA-dependent PM tension. The number of invadosomes in parental cells was found to increase significantly after RANKL treatment (Fig. 5, A and B). While ezrin-expressing cells showed similar behavior to parental cells, RANKL-induced invadosome formation was significantly suppressed in MA-ezrin and iMC-linker cells (Fig. 5, A and B). Conversely, knockdown of ezrin or ERM resulted in an increase in invadosome formation, with the number of mononuclear cells exhibiting invadosomes significantly increased as early as 60 h after RANKL treatment (Fig. 5, C and D). These findings indicate that MCA-dependent PM tension acts as a negative regulator at

the onset of cell–cell fusion by counteracting invadosome formation.

The observation that RANKL-induced reduction of PM tension promotes invadosome formation indicates the potential involvement of BAR proteins in this process. Knockdown of either Baiap2l1/IRTKS or Fnbp1/FBP17, which resulted in suppression of cell–cell fusion (Fig. 4 A), also led to a reduction in invadosome formation (Fig. 5, E and F). Consequently, a reduction in PM tension by ERM knockdown recruited both BAR proteins to RANKL-induced invadosomes (Fig. 5, G and H). These results suggest that RANKL-induced PM tension reduction promotes invadosome formation via BAR proteins, including Baiap2l1/IRTKS and Fnbp1/FBP17, thereby promoting cell–cell fusion.

The present study demonstrates that upon RANKL treatment to induce osteoclast differentiation, RAW 264.7 macrophages undergo a morphological transformation, shifting from a spindle-like cell morphology to a flat, spreading cell shape with a higher degree of circularity. The observation of successful fusion preferentially in cells with a well-spread morphology suggests a possible correlation between the apparent decrease in MCA and cell–cell fusion. Optical tweezer analysis directly demonstrated that the MCA was indeed reduced in RAW macrophages when treated with RANKL. Furthermore, osteoclast fusion was almost completely prevented when MCA was artificially increased by the expression of MA-ezrin or iMC-linker. These results collectively indicate an inverse correlation between MCA and cell–cell fusion efficiency. We expect that our conclusions will be evaluated in more detail in the future as other methods to directly manipulate membrane mechanics become available.

It was observed that ezrin expression in RAW macrophages decreased significantly 60 h after RANKL treatment (Fig. 3, A and B). The decrease in ezrin expression correlates with the decrease in MCA observed by optical tweezers (Fig. 2 B and Fig. 3 B). Interestingly, a recent study has reported that ezrin is also downregulated in differentiation to spreading macrophages (Verdys et al., 2024), consistent with our study. These findings suggest that ezrin may have a major role in the regulation of MCA, at least in fusion-competent macrophages. The RANKL signaling pathway is known to induce changes in a wide range of gene expression through the action of master transcriptional regulators, including NF-κB and NFATc1 (Ishida et al., 2002; Takayanagi et al., 2002). However, the precise function of these transcription factors in regulating ezrin expression remains unclear. Our in silico study indicates that a transcription factor, Blimp1, which acts as a negative regulator downstream of NFATc1, can bind to multiple *Ezr* promoter regions. The depletion of Blimp1 partially suppressed the RANKL-induced reduction of ezrin expression but did not completely overcome the effects of RANKL treatment (Fig. S3, K and L). Therefore, further investigation is required to elucidate the regulation of ezrin expression by additional transcription factors.

This study has shown that two groups of BAR family proteins, known to possess membrane-bending activity, are essential for osteoclast fusion. Baiap2l1/IRTKS, together with Baiap2/IRSp53 and Mtss1, form the "I-BAR" subfamily, which is known to induce and maintain various types of actin-dependent membrane

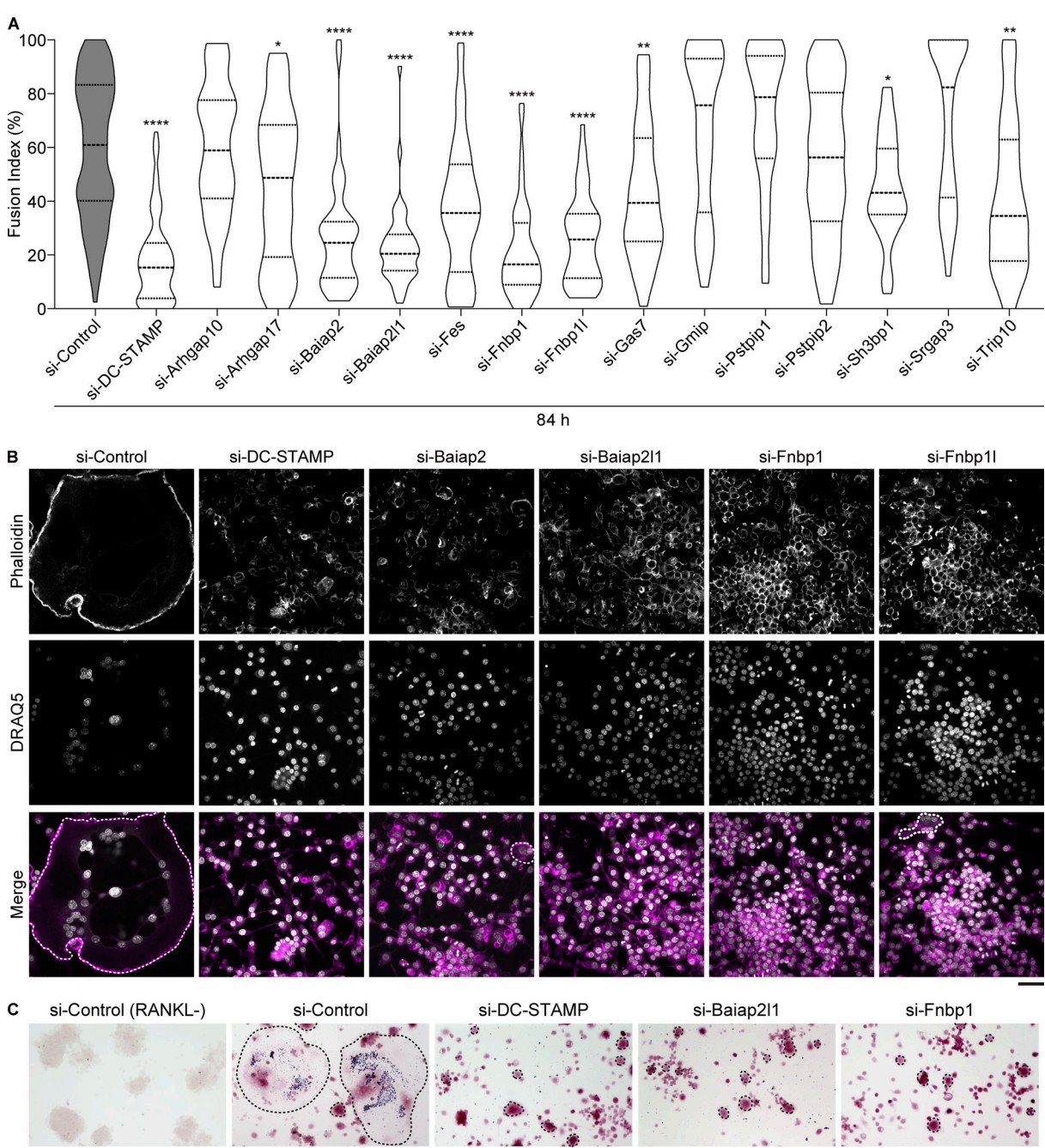

Figure 4. **Knockdown of Baiap2, Baiap2l1, and the TOCA family proteins inhibits cell–cell fusion by RANKL-induced osteoclastogenesis. (A)** The fusion index after treatment with the indicated RNAi targeting BAR proteins. Dotted lines in violin plots show median and quantiles. The total number of fields of view analyzed was as follows: n = 85 (si-Control), n = 80 (si-DC-STAMP), and n = 45 (si-BAR for each condition). P value obtained from one-way ANOVA with Dunnett's test. *P < 0.05; **P < 0.01; ****P < 0.0001. **(B)** Confocal images of siRNA knockdown cells treated with RANKL for 84 h, stained with phalloidin (magenta) and DRAQ5 (gray). Multinucleated cells are surrounded by dotted lines. Scale bar: 50 μm. **(C)** siRNA knockdown cells stained for TRAP activity. Multinucleated cells are surrounded by dotted lines. Note that siRNA knockdown cells treated with RANKL for 84 h indicate the activity of TRAP. Scale bar: 100 μm.

protrusion structures (Millard et al., 2007; Saarikangas et al., 2009). It has previously been shown that Baiap2l1 expression is increased by RANKL-induced osteoclast differentiation (Oikawa and Matsuo, 2012). The TOCA family of proteins contains F-BAR domains that allow deformation of the PM toward the cytosolic space. Fnbp1/FBP17 has been observed to form a dynamic dot-like structure surrounding the actin core of invadosomes in RAW macrophages (Tsujita et al., 2013). In addition, Fnbp1l/TOCA-1 has been shown to directly regulate filopodia formation (Lee et al., 2010). Our results, together with these observations, suggest that RANKL-induced PM tension reduction enables the assembly of membrane-bending BAR proteins, which in turn induce the protrusive membrane structures of invadosomes.

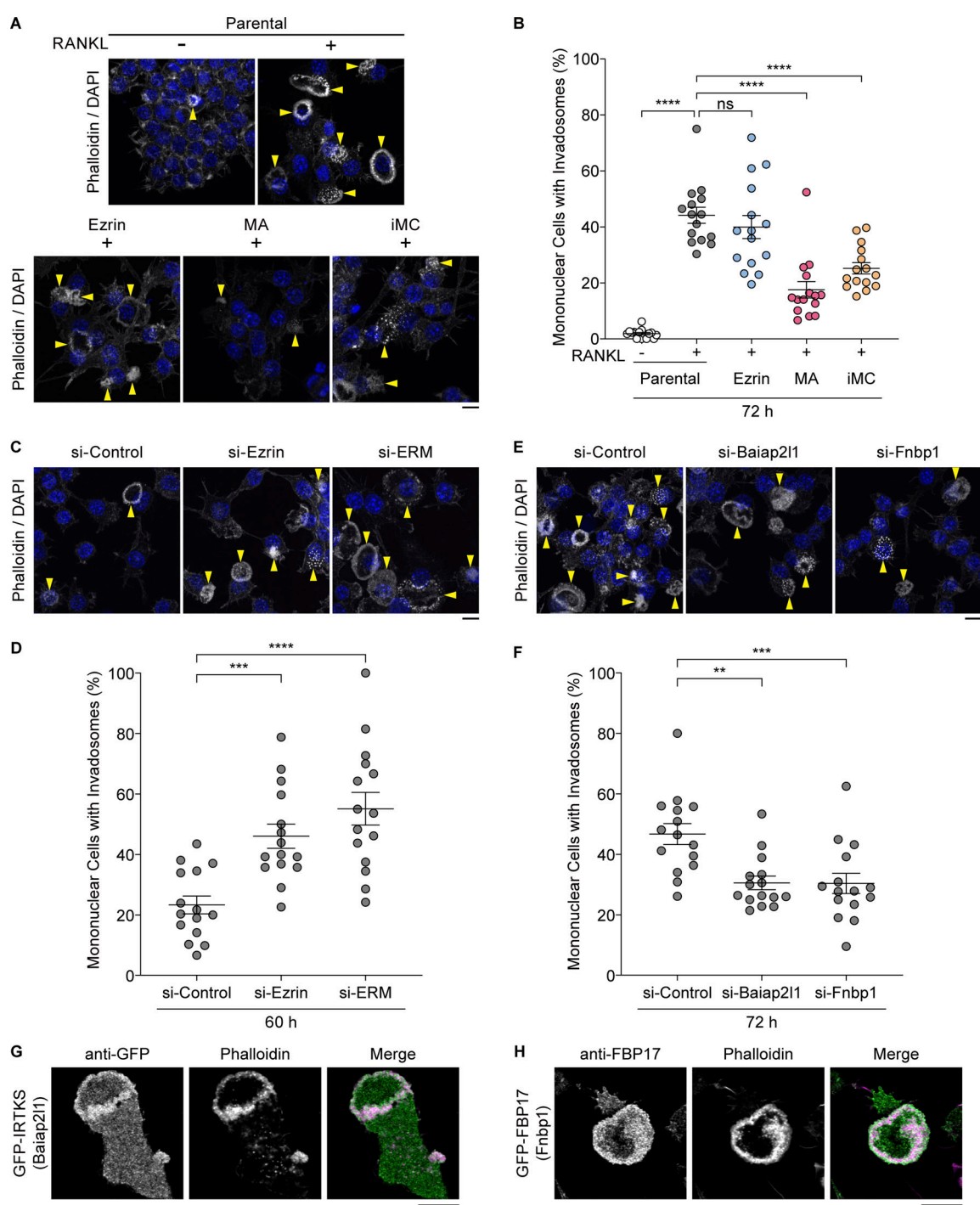

Figure 5. **Decreased MCA promotes invadosome formation for cell–cell fusion. (A)** Confocal images of parental cells and stable cell lines expressing ezrin, MA-ezrin (MA), and iMC-linker (iMC), without or with RANKL treatment for 72 h, stained with phalloidin (gray) and DAPI (blue). Arrowheads indicate mononuclear cells forming invadosomes. Scale bar: 10 μm. **(B)** Quantification of the percentage of mononuclear cells forming invadosomes in A. Mean ± SEM, *n* = 15 fields of view for each condition. P value obtained from one-way ANOVA with Dunnett's test. ns: not significant, ****P < 0.0001. **(C)** Confocal images of ezrin and ERM knockdown cells treated with RANKL for 60 h, stained with phalloidin (gray) and DAPI (blue). Arrowheads indicate mononuclear cells forming invadosomes. Scale bar: 10 μm. **(D)** Quantification of (C). Mean ± SEM, *n* = 15 fields of view for each condition. P value obtained from one-way ANOVA with Dunnett's test. ***P < 0.001; ****P < 0.0001. **(E)** Confocal images of Baiap2l1 and Fnbp1 knockdown cells treated with RANKL for 72 h, stained with phalloidin (gray) and DAPI (blue). Arrowheads indicate mononuclear cells forming invadosomes. Scale bar: 10 μm. **(F)** Quantification of (E). Mean ± SEM, *n* = 15 fields of view for each condition. P value obtained from one-way ANOVA with Dunnett's test. **P < 0.01; ***P < 0.001. **(G)** Confocal image of a stable cell line expressing GFP-IRTKS stained with anti-GFP antibody (green) and phalloidin (magenta) after ERM knockdown and RANKL treatment for 72 h. Scale bar: 10 μm. **(H)** Confocal image of a stable cell line expressing GFP-FBP17 stained with anti-FBP17 antibody (green) and phalloidin (magenta) after ERM knockdown and RANKL treatment for 72 h. Scale bar: 10 μm.

Previous studies have elucidated the relationship between cell–cell fusion and cortical tension, particularly in the context of *Drosophila* muscle development (Kim et al., 2015). In this process, the founder cells exhibit increased cortical tension, which is dependent on myosin II and spectrin. Subsequently, the myoblast extends actin-rich membrane protrusions that eventually lead to fusion with the founder cell (Shilagardi et al., 2013; Duan et al., 2018). Our study suggests that a symmetric reduction in PM tension is important for osteoclast fusion. Perhaps in osteoclast fusion, invasive membrane protrusions in the form of invadosomes of both fusing cells, rather than in a heterotypic manner, may play a key role in fusion processes. It has been shown that the EPS8–IRSp53 complex is degraded by the E3 ligase CUL3 KCTD10, resulting in the disruption of the cortical actin bundle, thereby allowing cell–cell fusion to occur (Rodríguez-Pérez et al., 2021), which is consistent with our observations. Our data provide important insights into the role of cell mechanics in the regulation of cell–cell fusion dynamics.

## Materials and methods

### Antibodies and materials

The following antibodies were used: anti–p-ERM (rabbit monoclonal, 3726; Cell signaling technology), anti-ezrin (rabbit polyclonal, 3145; Cell Signaling Technology), anti-radixin (rabbit monoclonal, 2636; Cell Signaling Technology), anti-moesin (rabbit polyclonal, 3146; Cell Signaling Technology), anti–β-actin (rabbit polyclonal, PM053; MBL), anti-NFATc1(7A6) (mouse monoclonal, sc-7294; Santa Cruz Biotechnology), anti-HA tag (mouse monoclonal, 2367; Cell Signaling Technology), anti-GFP (rabbit polyclonal, 598; MBL), and anti-FBP17 (rabbit polyclonal, home-made [Itoh et al., 2005]). HRP-conjugated secondary antibodies (anti-rabbit, 111-035-144; anti-mouse, 115-035-146) were purchased from Jackson ImmunoResearch. Alexa Fluor 488–conjugated secondary antibodies (anti-rabbit, A11034; anti-mouse, A11001), Alexa Fluor 568–conjugated secondary antibody (anti-rabbit, A11036), Alexa Fluor 568 Phalloidin (A12380), DAPI (D1306), and Alexa Fluor 488–conjugated wheat germ agglutinin (WGA488) (W11261) were purchased from Invitrogen. DRAQ5 (424101) was obtained from BioLegend, and MG132 (474790) from Merck Millipore.

Recombinant mouse RANKL was purchased from R&D Systems (462-TEC-010) or prepared by affinity purification of bacterially expressed GST-RANKL (aa 158–316 of mouse RANKL), followed by tag removal (PreScission protease; Cytiva) and further purification by cation exchange column (Enrich S; Bio-Rad).

### Plasmid construction

The cDNAs encoding ezrin and MA-ezrin were generated as previously described (Tsujita et al., 2021). Briefly, human ezrin was inserted into pQCXIN-HA retroviral vector, which was modified pQCXIN retroviral vector (Clontech) by introducing an HA tag to the C terminus. $Lyn_{10}$-ezrinT567E (MA-ezrin) was constructed by the fusion of a myristoylation motif (MGCIKSKRKD: $Lyn_{10}$) derived from Lyn tyrosine kinase as the PM-targeting signal to the N terminus of human ezrin, followed by the generation of the T567E point mutation using PCR primers, and inserted into pQCXIN-HA retroviral vector. iMC-linker was constructed by the fusion of $Lyn_{10}$ to the N terminus of the F-actin–binding domain of human utrophin (aa 1–261) and inserted into pQCXIN-HA retroviral vector (Bergert et al., 2021).

The cDNAs encoding human BAIAP2L1 and human FNBP1 were cloned in our laboratory and subcloned into the pRetroX-TetOne-Puro retroviral vector (Takara) with the cDNA of monomeric EGFP (A206K) fused to the N terminus of BAIAP2L1 or FNBP1 to produce GFP-IRTKS or GFP-FBP17. A linker sequence of 15 aa (GGGGS × 3) was inserted between GFP and the target protein. Transgene expression was assessed by western blot analysis (Fig. S2 A) and confocal microscopy.

### Cell culture

RAW 264.7 cells, a mouse macrophage cell line, were purchased from the American Type Culture Collection. RAW 264.7 cells were cultured in DMEM (08458-16; Nacalai Tesque) supplemented with 10% FBS (F7524; Sigma-Aldrich) under 5% $CO_2$ atmosphere at 37°C. For the cell–cell fusion experiments, RAW 264.7 cells were inoculated in a 35-mm–ϕ glass bottom dish (3911-035; Iwaki) or a 6-well plate at a density of $2.8 \times 10^4$/dish or $2.8 \times 10^4$/well and added 50 ng/ml RANKL after 12 h. The cells were further cultured for 12–84 h depending on the experiments.

GP2–293 cell line was purchased from Thermo Fisher Scientific. The cells were maintained in DMEM (08458-16; Nacalai Tesque) supplemented with 10% FBS (F7524 or 172012; Sigma-Aldrich) under 5% $CO_2$ atmosphere at 37°C and used for retroviral packaging.

### Retroviral gene transduction

To obtain stable cell lines expressing ezrin, MA-ezrin, iMC linker, GFP-IRTKS, and GFP-FBP17, retroviruses were produced by co-transfection to GP2–293 cells with pQCXIN or pRetroX-TetOne-Puro constructs as above and pVSV-G (Takara) using Xfect Transfection Reagent (631318; Takara) as per the manufacturer's protocols. RAW 264.7 cells were infected with retroviruses, followed by selection with 800 µg/ml G418 (09380-44; Nacalai Tesque) for pQCXIN constructs or 5 µg/ml puromycin (ant-pr-1; InvivoGen) for pRetroX-TetOne-Puro constructs.

### RNAi and transfection

For knockdown experiments, SMARTpool-ON–TARGETplus siRNAs (Horizon) against mouse genes were used Ezr (L-046568-01-0005), Rdx (L-047230-01-0005), Msn (L-044428-01-0005), Arhgap10 (L-044764-01), Arhgap17 (L-040551-02), Baiap2 (L-046696-01), Baiap2l1 (L-041646-01), Fes (L-043381-00), Fnbp1 (L-062539-01), Fnbp1l (L-054569-01), Gas7 (L-045653-01), Gmip (L-055099-01), Pstpip1 (L-043941-01-0005), Pstpip2 (L-062910-01-0005), Sh3bp1 (L-044982-01), Srgap3 (L-058941-01), Trip10 (L-052230-01), Dcstamp (L-049310-01-0005), and Prdm1 (L-043069-01-0005). ON-TARGETplus Nontargeting siRNA Pool (D-001810-10-05) was used as a control siRNA.

RAW 264.7 cells were inoculated in a 35-mm–φ glass bottom dish or a 6-well plate at a density of $2.8 \times 10^4$/dish or $2.8 \times 10^4$/well. After 12 h, the cells were transfected 12.5 nM siRNA with Lipofectamine RNAi MAX (13778075; Invitrogen) and supplemented 50 ng/ml RANKL. The cells were further cultured for 60–84 h until the analysis.

## Quantitative real-time PCR

Total RNA was isolated from the samples using TRIzol reagent (15596-018; Thermo Fisher Scientific) or Sepasol-RNA I Super G (09379-55; Nacalai Tesque), and portions (0.1–1 µg) of the RNA were subjected to reverse transcription with polymerase (FSQ-301; TOYOBO). Primers for *Ezr*, *Rdx*, *Msn*, *Nfatc1*, *Fnbp1*, *Baiap2l1*, and *Gapdh* were obtained from Eurofins Genomics. Quantitative real-time PCR was performed on StepOnePlus Real Time PCR System (Applied Biosystems) using Thunderbird Next SYBR qPCR Mix (QPX-201; TOYOBO). Results were normalized to the abundance of mRNA derived from the housekeeping gene *Gapdh* and quantified using the $\Delta\Delta C_T$ method. The primers used in the experiment were as follows (forward and reverse, respectively): *Ezr*, 5′-GTACAGCCGAATAGCCGAGG-3′ and 5′-AGCATAGCACTGTCCTTGAGC-3′; *Rdx*, 5′-AGCTGTGGCTAGGTGTTGATG-3′ and 5′-CAGACGAGGTGCATAGAAGAC-3′; *Msn*, 5′-TGGATGCAGAGCTGGAGTTTG-3′ and 5′-GAGAATGCCCTCCTTCACTTG-3′; *Nfatc1*, 5′-GGCCGAGGAAGAACACTACAGTTATG-3′ and 5′-TGGAAAAACTGGCCGCTGCCATG-3′; *Fnbp1*, 5′-AAGGCACGACAACAAGCTC-3′ and 5′-ATACGCACAATCCGCCTCTC-3′; *Baiap2l1*, 5′-CTCACGGAGAACACGTACCG-3′ and 5′-TGGCCTAGCTCTGTTGACAC-3′; and *Gapdh*, 5′-CAGGTTGTCTCCTGCGACTT-3′ and 5′-AGCCGTATTCATTGTCATACCAGG-3′.

## Western blotting and analysis

RAW 264.7 cells were inoculated in a 6-well plate at a density of $2.8 \times 10^4$/well. After 12 h, the cells were supplemented with 50 ng/ml RANKL. At this time, siRNA transfection was performed as needed. For proteolysis inhibition experiments, 0.2 µM protease inhibitor MG132 (474790; Merck Millipore) was supplemented 60 h after the addition of RANKL. The cells were further cultured until the analysis. The cells were extracted with 120–150 µl of 2× Laemmli SDS sample buffer (0.125 M Tris-HCl buffer, 10% 2-mercaptoethanol, 4% SDS, 10% sucrose, and 0.01% bromophenol blue), sonicated, and incubated for 10 min at 95°C. After centrifugation, the concentration of total protein was determined by Pierce 660 nm Protein Assay Reagent (22660; Thermo Fisher Scientific) and Ionic Detergent Compatibility Reagent (22663; Thermo Fisher Scientific). The proteins were electrophoresed in SDS-PAGE gels (197–15011; Fujifilm Wako), transferred to polyvinyl difluoride membrane (IB24001; iBlot2 PVDF Regular Stacks, Invitrogen) by iBlot2 (Invitrogen), and blocked with 5% nonfat skim milk (31149-75; Nacalai Tesque) in TBS containing 0.1% Tween20 (TBS-T, pH 7.4, 207–18061; Fujifilm Wako) for 60 min at RT. Then, membranes were probed with primary antibodies diluted in Solution 1 (NKB-101; Can Get Signal, TOYOBO) overnight at 4°C. Primary antibodies and dilutions we used for western blotting were as follows: anti-ezrin (1:1,000, 3145S; Cell Signaling Technology), anti–p-ERM (1:1,000, 3726S; Cell Signaling Technology), anti-radixin (1:1,000, 

2636; Cell Signaling Technology), anti-moesin (1:1,000, 3146S; Cell Signaling Technology), anti-NFATc1(7A6) (1:1,000, sc-7294; Santa Cruz Biotechnology), and anti-HA tag (1:1,000, 2367; Cell Signaling Technology). Only for anti–β-actin (1:5,000, PM053; MBL), membranes were probed for 60 min at RT. After washing three times with TBS-T, the membranes were incubated with HRP-conjugated secondary antibodies (1:5,000, anti-rabbit, 111-035-144; anti-mouse, 115-035-146; Jackson ImmunoResearch) diluted in Solution 2 (NKB-101; Can Get Signal, TOYOBO) for 60 min at RT. The bands of the samples were visualized by chemiluminescence (292-69903; ImmunoStar LD, Fujifilm Wako, or 07880-70; Chemi-Lumi One L, Nacalai Tesque), and then quantified the intensity of the bands using Fiji software (Schindelin et al., 2012). The expression levels of NFATc1 and ezrin (after Blimp1 knockdown) were quantified and normalized to the β-actin intensities (Fig. S2 C and Fig. S3 L). The gray value of the target band was measured by drawing a rectangle around it and subtracting the average gray value of the upper and lower regions of the target band as a background.

## Time-lapse microscopy and analysis of cell morphology

Time-lapse observation for RAW 264.7 cells was performed from 48 h after RANKL addition under 5% $CO_2$ atmosphere at 37°C in the $CO_2$ stage chamber (Tokai Hit), using Keyence BZ-X710 microscope equipped with a Nikon Plan Fluor 10× (NA 0.30) objective lens. Images were captured every 30 min for 48 h (until 96 h after the addition of RANKL). Cell morphology was semiautomatically detected using the open-source deep learning software Cellpose (Stringer et al., 2021) and analyzed using Fiji software (Schindelin et al., 2012). "Time 0" was defined as the time of cell–cell fusion for RANKL-treated fused cells and an arbitrary time for non–RANKL-treated cells and RANKL-treated non-fused cells (Fig. 1, E–H).

## Immunofluorescence and confocal microscopy

For fusion index analysis, cells were fixed with 4% PFA in PBS (09154-85; Nacalai Tesque) for 15 min, permeabilized with 0.4% Triton X-100 in PBS for 15 min, and incubated with Alexa Fluor 568 Phalloidin (1:500, A12380; Invitrogen) for 60 min at RT, followed by staining with DAPI (1:10,000, D1306; Invitrogen) or DRAQ5 (1:10,000, 424101; BioLegend) for 10 min at RT.

For quantification of p-ERM beneath the PM, cells were fixed with 4% PFA in PBS for 15 min and incubated with WGA488 (5 µg/ml, W11261; Invitrogen) for 10 min at RT to visualize the PM. Then, the cells were permeabilized with 0.4% Triton X-100 in PBS for 15 min, blocked with 5% goat serum (G9023; Sigma-Aldrich) in PBS for 60 min, and incubated with anti–p-ERM primary antibody (1:200, 3726S; Cell Signaling) overnight at 4°C. Then, the cells were incubated with Alexa Fluor 568–conjugated goat anti-rabbit IgG secondary antibody (1:500, A11036; Invitrogen) for 60 min at RT and DRAQ5 (1:10,000, 424101; BioLegend) for 10 min at RT.

For immunostaining of NFATc1, after fixation with 4% PFA, permeabilization with 0.4% Triton X-100, and blocking with 5% goat serum, cells were incubated with anti-NFATc1(7A6) primary antibody (1:100, sc-7294; Santa Cruz Biotechnology) diluted by Can Get Signal immunostain Immunoreaction Enhancer

Solution B (NKB-401; TOYOBO) overnight at 4°C. Then, the cells were incubated with Alexa Fluor 488–conjugated goat anti-mouse IgG secondary antibody (1:500, A11001; Invitrogen), Alexa Fluor 568 Phalloidin (1:200, A12380; Invitrogen), and DAPI (1:10,000, D1306; Invitrogen) for 60 min at RT.

For observation of invadosomes, phalloidin and DAPI staining were performed as above.

To visualize Baiap2l1/IRTKS and Fnbp1/FBP17 with invadosomes, cells were treated with 50 ng/ml RANKL, si-ERM, and 1 μg/ml doxycycline (8634-1; Clontech) for 72 h, and immunostained with anti-GFP (1:200, 598; MBL) or anti-FBP17 (1:100; home-made) antibodies for 3 h at RT, followed by Alexa Fluor 488–conjugated secondary antibody (1:500, A11034; Invitrogen) and Alexa Fluor 568 Phalloidin (1:200, A12380; Invitrogen) for 60 min at RT.

Fluorescence images were captured using a confocal microscopy system FluoView 1000-D (Olympus) equipped with Olympus UPlan SApo 100× (NA 1.40) and 40× (NA 0.90) objective lenses, BC43 (Andor) with Nikon Plan Apo 40× (NA 0.95) and 60× (NA 1.42) objective lenses, and Dragonfly (Andor) with Nikon Plan Apo 100× (NA 1.49) objective lens.

### Quantitative analysis based on the confocal images

For all of the analyses in this paper, phalloidin was used to confirm cell boundaries, DAPI and DRAQ5 were used to determine the number and region of nuclei, and WGA488 signal as the PM marker.

To measure the fusion ability as the fusion index, we pooled only the cells that were in direct contact on the basal surface based on the confocal images, and the cells with at least three nuclei were determined to be multinucleated cells to eliminate binucleated cells undergoing cytokinesis. The number of nuclei was counted manually, and the ratio of the number of nuclei in a multinucleated cell to the total number of nuclei in the field of view (317.3 μm squared for FV1000, Olympus and 318.2 × 310.6 μm for BC43; Andor) was defined as the fusion index (Shilagardi et al., 2013). The fusion index was obtained from 30 to 85 fields of view per condition.

For the nucleus-to-cytoplasm ratio of NFATc1, DAPI, and phalloidin images were segmented using Cellpose (Stringer et al., 2021) with the nuclear region and cellular region, and the difference between the nuclear region and cellular region was defined as the cytoplasmic region. The average signal intensity of NFATc1 was calculated for the nuclear region and the cytoplasmic region to obtain the nucleus-to-cytoplasm ratio for each mononuclear cell, using Fiji software (Schindelin et al., 2012).

To calculate the fluorescence intensity of p-ERM beneath the PM, the membrane region of the cell was selected manually using the WGA488 signal as the PM marker, with a two-pixel width (about 400 nm) using Fiji software's selection brush tool. Based on this membrane region, we measured the average signal intensity of p-ERM for each mononuclear cell.

Invadosomes were defined as the assembly of actin dots of more than 20 per cell, or an actin cluster with a diameter >5 μm, including the actin ring. The ratio of cells forming invadosomes was defined as the percentage of mononuclear cells that form invadosomes relative to the total number of mononuclear cells in the field of view and was measured manually.

### Tether force measurement utilizing optical tweezers

Tether force measurements were performed as previously described with modifications (Tsujita et al., 2021). Briefly, RAW 264.7 cells treated with RANKL for 60 h were inoculated in a 35-mm–ϕ glass bottom dish (FD35-100; WPI) to disperse each cell and further cultured for 24 h. The tether force of the PM was measured using an optical tweezers (NanoTrackerTM 2, JPK Instruments) outfitted with a near-infrared laser (3 W, 1,064 nm) on an Olympus IX-73 with a 60× (NA 1.2) objective lens. Silica microspheres (1.5-μm–ϕ, 24327-15, Polysciences) were coated with concanavalin A (C5275; Sigma-Aldrich) at 1 mg/ml for 60 min at RT and then added to the cell culture medium supplemented with 40 mM HEPES (pH 7.5, 15630-080; Gibco). The experiments were carried out at 37°C within 30 min, and measured tether force ($F$) of RANKL-treated and non-treated mononuclear cells on the lateral sides of cells. Tether force can be calculated according to Hooke's law,

$$F = k \times \Delta x \qquad (1)$$

where $k$ is the stiffness of the trap and $\Delta x$ is the displacement of the bead from the trap center. Trap stiffness ($k$, typically ~0.15 pN/nm) was calibrated for each experiment by a power spectrum analysis (Berg-Sørensen and Flyvbjerg, 2004). A single bead trapped by optical tweezers was brought into contact with the PM of a mononuclear cell for 500 ms and then pulled away from the cell at the rate of 1 μm/s to form a membrane tether (5 μm length). The displacement of the bead in the trap center ($\Delta x$) was detected by a quadrant photodiode detector with <1 nm spatial resolution. Data were analyzed using the JPK data processing software. Because cells with low PM tension, such as RANKL-treated cells, were more frequently to form double tethers, which exhibited a twofold increase in the tether force, we confirmed whether or not one tether was formed and excluded data from double tethers.

### TRAP staining

For TRAP staining, differentiated RAW 264.7 cells were fixed with 4% PFA in PBS for 15 min at RT and permeated with pre-cold ethanol/acetone (50:50 vol/vol) for 1 min at –20°C. Then, TRAP activity was detected by tartrate with the use of a kit (TRAP/ALP Stain Kit, 294–67001; Fujifilm Wako) and captured using Keyence BZ-X710 microscope equipped with a Nikon Plan Fluor 10× (NA 0.30) objective lens.

### Software and statistical analysis

Statistical analyses were carried out with GraphPad Prism and Origin Pro. Statistical significance was determined using one-way ANOVA with Tukey's test or Dunnett's test for multiple comparisons and two-tailed Student's $t$ test. $P < 0.05$ was considered statistically significant. The sample sizes for each experiment are stated in the figure legends.

### Online supplemental material

Fig. S1 shows that the cell–cell fusion is initiated at 48 h after RANKL treatment with morphological changes. Fig. S2 shows the establishment of cell lines expressing ezrin, MA-ezrin, and iMC-linker, and the expression and localization confirmation of

NFATc1, a master transcription factor, and TRAP, an osteoclast differentiation marker. Fig. S3 shows that ezrin is the only ERM family protein whose expression is reduced by RANKL treatment, and its expression is regulated at the transcriptional level. Video 1 shows the time-lapse imaging of cell–cell fusion between mononuclear cells, related to Fig. 1 B. Video 2 shows time-lapse imaging of ezrin and ERM proteins knockdown cells, related to Fig. 3, F and G.

### Data availability

All data supporting the findings of this study are available from the corresponding authors upon reasonable request.

## Acknowledgments

This work was supported by the Japan Society for the Promotion of Science KAKENHI (JP23K23838 to T. Itoh, JP23K27380 to K. Tsujita, JP24K09447 to T. Oikawa, JP21K06078 to K. Takano, and JP23K19352 to Y.L. Nemoto), ONO Medical Research Foundation (to T. Itoh), The Mitsubishi Foundation (to T. Itoh), Hyogo Science and Technology Association (to T. Itoh), SGH Cancer Research Grant (to T. Oikawa), and Biosignal Research Center Core Facility Research Funding (201005 to K. Takano). Open Access funding provided by Kobe University.

Author contributions: Y. Wan: data curation, formal analysis, investigation, methodology, validation, visualization, writing—original draft, review and editing. Y.L. Nemoto: data curation, formal analysis, funding acquisition, investigation, methodology, software, validation, visualization, and writing—original draft, review, and editing. T. Oikawa: funding acquisition, investigation, methodology, resources, and writing—review and editing. K. Takano: conceptualization, funding acquisition, and writing—review and editing. T.K. Fujiwara: investigation. K. Tsujita: conceptualization, data curation, formal analysis, funding acquisition, investigation, methodology, project administration, resources, supervision, validation, visualization, and writing—original draft, review, and editing. T. Itoh: conceptualization, data curation, formal analysis, funding acquisition, investigation, methodology, project administration, resources, software, supervision, validation, visualization, and writing—original draft, review, and editing.

Disclosures: The authors declare no competing interests exist.

Submitted: 5 November 2024

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

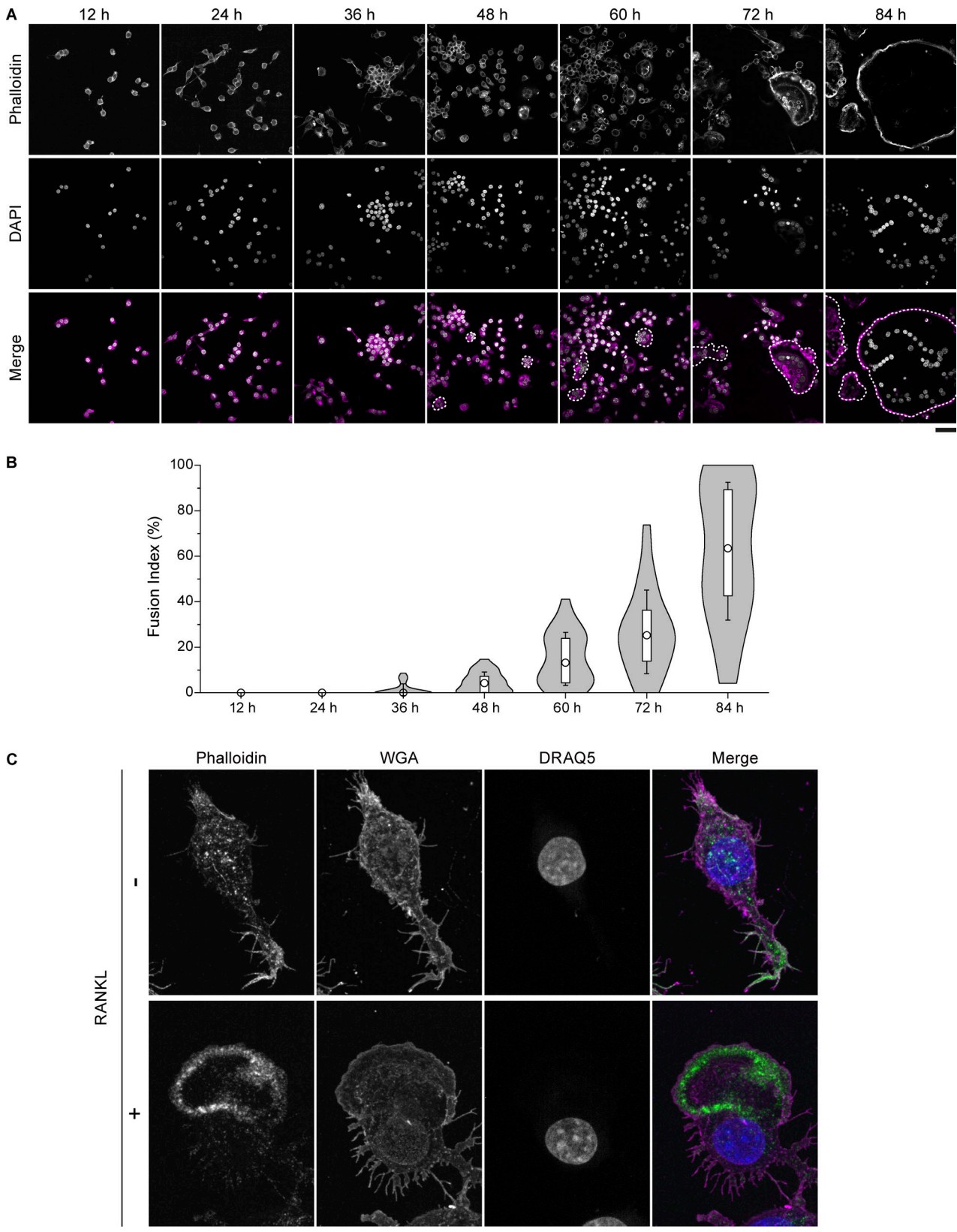

Figure S1.   **Cell–cell fusion is initiated at 48 h after RANKL treatment with morphological changes. (A)** Confocal images of the time sequence of RAW 264.7 cells after RANKL addition, stained with phalloidin (magenta) and DAPI (gray). Multinucleated cells are surrounded by dotted lines. Scale bar: 50 μm. **(B)** Quantification of fusion index in A. For box plots inside violin plots, circles, boxes, and whiskers indicate median values, interquartile range (25–75%), and SD, respectively. n = 30 fields of view for each time point. **(C)** Representative confocal images of mononuclear cells without or with RANKL treatment for 84 h, stained with phalloidin (green), WGA (magenta), and DRAQ5 (blue). Scale bar: 10 μm. WGA, wheat germ agglutinin.

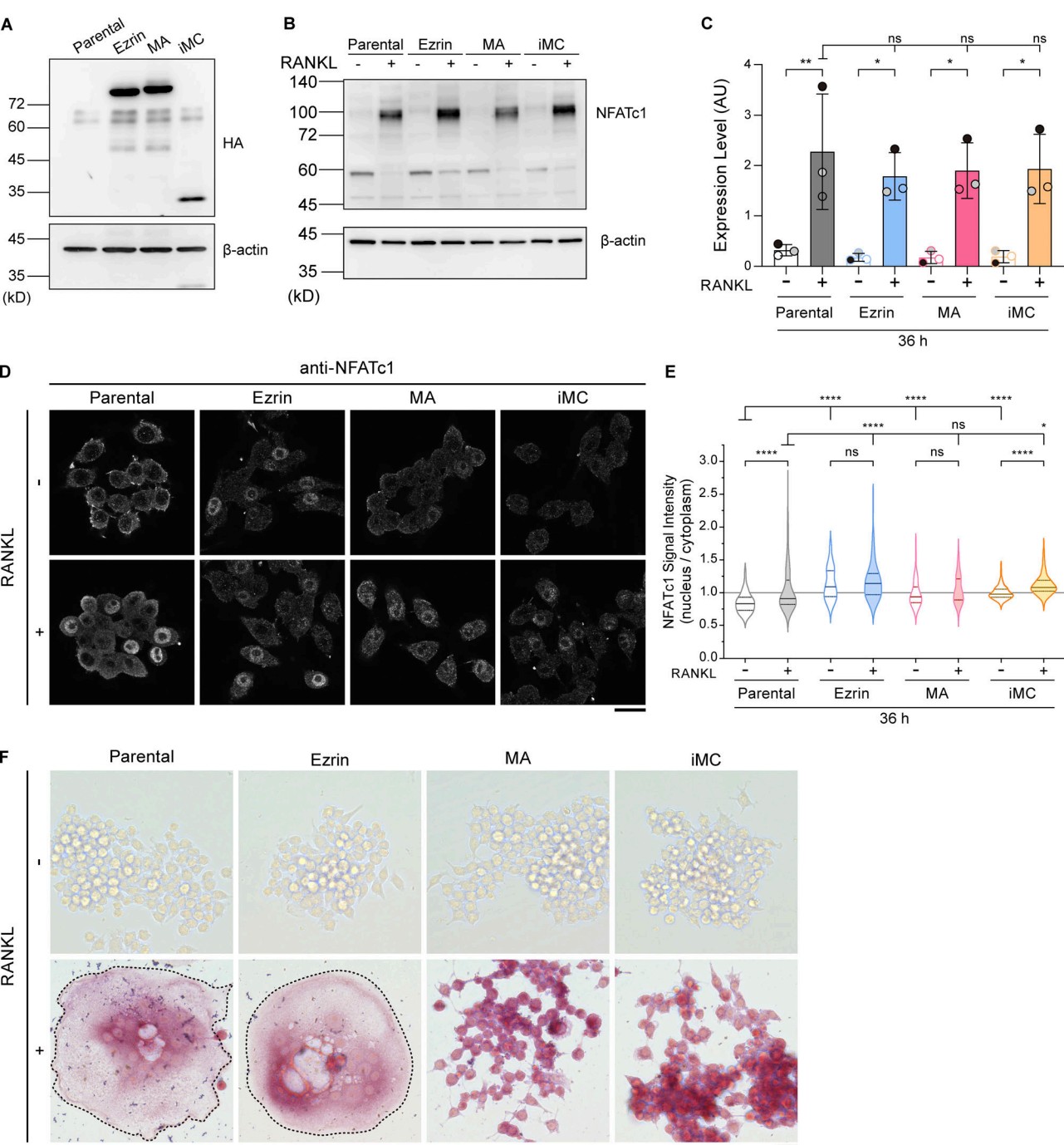

Figure S2. **Establishment of cell lines expressing ezrin, MA-ezrin, and iMC-linker, and the expression and localization confirmation of NFATc1, a master transcription factor, and TRAP, an osteoclast differentiation marker. (A)** Representative western blot of parental cells and stable cell lines overexpressing HA-tagged ezrin, MA-ezrin (MA), and iMC-linker (iMC) without RANKL treatment, using anti-HA tag and anti–β-actin antibodies. **(B)** Representative western blot of parental cells and stable cell lines above without or with RANKL treatment for 36 h, using anti-NFATc1 and anti–β-actin antibodies. **(C)** Quantification of B. Mean ± SEM, *n* = 3 experiments. Note that overexpression of these exogenous genes does not inhibit the induction of NFATc1 expression induced by RANKL. P value obtained from one-way ANOVA with Tukey's test. ns: not significant; *P < 0.05; **P < 0.01. **(D)** Confocal images of the four cell lines above without or with RANKL treatment for 36 h, stained with anti-NFATc1 antibody. Scale bar: 20 μm. **(E)** The nucleus-to-cytoplasm ratio of NFATc1 signal intensity in mononuclear cells. Solid and dotted lines in violin plots show median and quantiles. The total number of cells analyzed was as follows: *n* = 224 (parental, RANKL−), 197 (parental, RANKL+), 153 (ezrin, RANKL−), 200 (ezrin, RANKL+), 141 (MA, RANKL−), 117 (MA, RANKL+), 288 (iMC, RANKL−), and 304 (iMC, RANKL+). P value obtained from one-way ANOVA with Tukey's test. ns: not significant; *P < 0.05; ****P < 0.0001. Note that the nucleus-to-cytoplasm ratios of these stable cell lines tend to be higher compared with parental cells; however, they do not inhibit the nuclear transport of NFATc1. **(F)** The four cell lines stained for TRAP activity. Multinucleated cells are surrounded by dotted lines. Note that all cell lines indicate the activity of TRAP after RANKL treatment for 84 h. Scale bar: 200 μm. Source data are available for this figure: SourceData FS2.

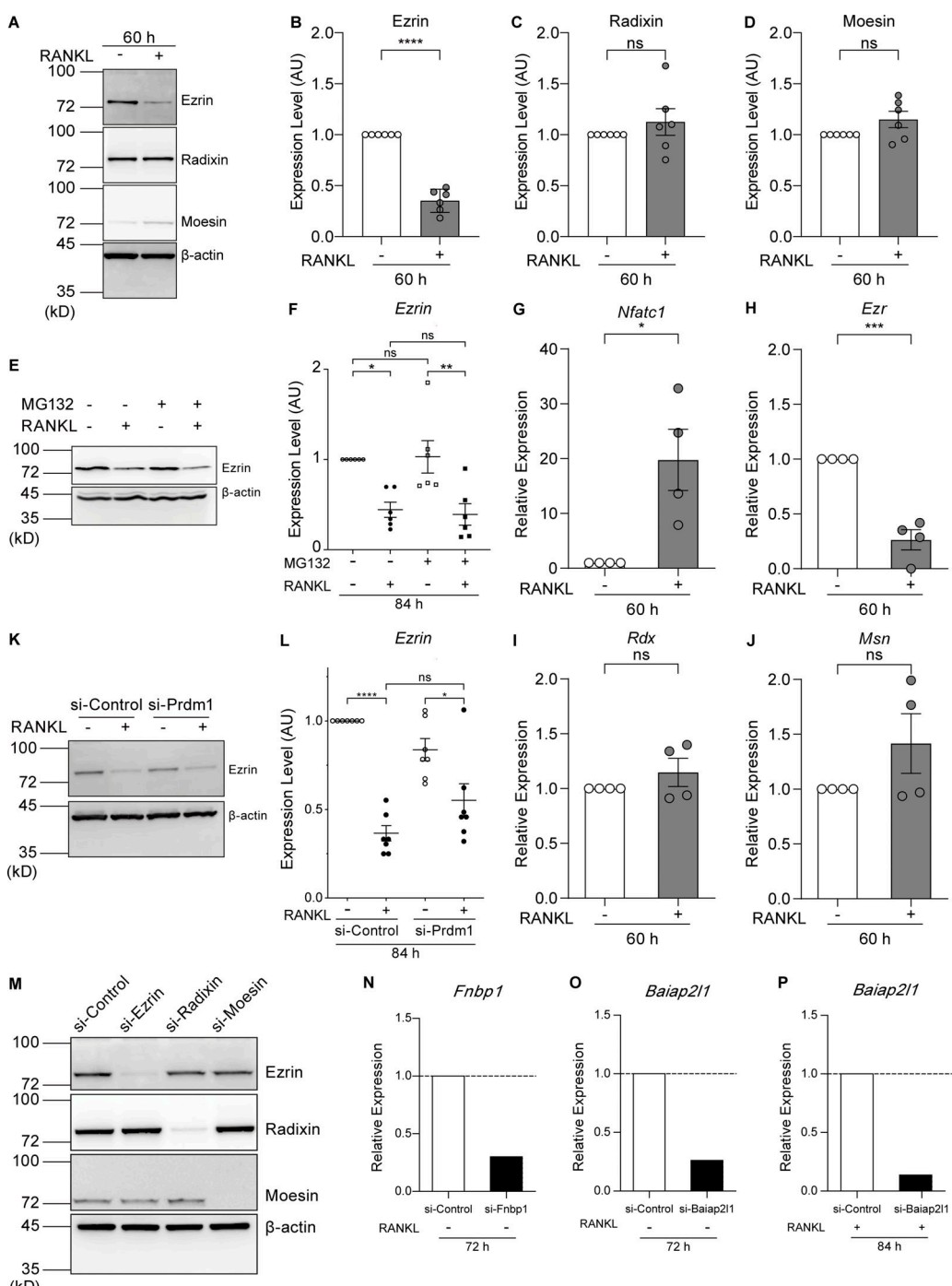

Figure S3. **Ezrin is the only ERM family protein whose expression is reduced by RANKL treatment, and its expression is regulated at the transcriptional level. (A)** Western blot analysis of cell lysates without or treated with RANKL for 60 h using anti-ezrin, anti-radixin, anti-moesin, and anti–β-actin antibodies. **(B–D)** Quantification of endogenous ezrin (B), radixin (C), and moesin (D) based on (A). Mean ± SEM, n = 6 experiments. P value obtained from two-tailed Student's t test. ns: not significant; ****P < 0.0001. **(E)** Western blot analysis of cell lysates without or treated with RANKL and the proteasome inhibitor MG132 for 84 h using anti-ezrin and anti–β-actin antibodies. **(F)** Quantification of endogenous ezrin based on E. The values of each band were measured and normalized to cells not treated with RANKL and MG132. Mean ± SEM, n = 6 experiments. P value obtained from one-way ANOVA with Tukey's test. ns: not significant; *P < 0.05; **P < 0.01. **(G–J)** qPCR analysis of the relative expression of *Nfatc1* (G), *Ezr* (H), *Rdx* (I), and *Msn* (J). mRNA was extracted after 60 h of RANKL treatment and normalized to the expression of *Gapdh* mRNA. Mean ± SEM, n = 4 experiments. P value obtained from two-tailed Student's t test. ns: not significant; *P < 0.05; ***P < 0.001. **(K)** Western blot analysis of Blimp1 knockdown cell lysates without or treated with RANKL for 84 h using anti-ezrin and anti–β-actin antibodies. **(L)** Quantification of endogenous ezrin based on K. The values of each band were measured and normalized to cells not treated with RANKL of si-Control group. Mean ± SEM, n = 7 experiments. P value obtained from one-way ANOVA with Tukey's test. ns: not significant; *P < 0.05; ****P < 0.0001. **(M)** Representative western blot of siRNA knockdown cells with RANKL treatments for 70 h, using anti-ezrin, anti-radixin, anti-moesin, and anti–β-actin antibodies. **(N–P)** qPCR analysis of the relative expression of *Fnbp1* (N) and *Baiap2l1* (72 h, O; 84 h, P) mRNAs, normalized by that of *Gapdh* mRNA. Source data are available for this figure: SourceData FS3.

Video 1. **Time-lapse imaging of cell–cell fusion between mononuclear cells, related to** Fig. 1 B. Mononuclear and binuclear cells analyzed for morphological changes are indicated with arrowheads (mononuclear cells in magenta and green, and a binuclear cell in blue). Image acquisition began 48 h after RANKL addition and was taken every 30 min. The number in the bottom left indicates the frame number.

Video 2. **Time-lapse imaging of ezrin and ERM proteins knockdown cells, related to** Fig. 3, F and G. Compared with the control, knockdown of ezrin and ERM proteins accelerates cell–cell fusion and the appearance of multinucleated giant cells earlier. Image acquisition began 48 h after siRNA and RANKL treatment and was taken every 30 min for 24 h. Displayed time indicates the time elapsed since RANKL addition.

