## [Peer Review File · The Journal of Cell Biology]

Mechanical control of osteoclast fusion by membrane-cortex attachment and BAR proteins

Yumeng Wan, Yuri Nemoto, Tsukasa Oikawa, Kazunori Takano, Takahiro Fujiwara, Kazuya Tsujita, and Toshiki Itoh

Corresponding Author(s): Toshiki Itoh, Kobe University and Kazuya Tsujita, Kobe University

Review Timeline:

Submission Date:	2024-11-05
Editorial Decision:	2025-01-08
Revision Received:	2025-03-13
Editorial Decision:	2025-03-31
Revision Received:	2025-04-01

Monitoring Editor: Pekka Lappalainen

Scientific Editor: Andrea Marat

Transaction Report:

DOI: <https://doi.org/10.1083/jcb.202411024>

January 8, 2025

Re: JCB manuscript #202411024

Toshiki Itoh
Kobe University

Dear Prof. Itoh,

Thank you for submitting your manuscript entitled "Mechanical control of osteoclast fusion by membrane-cortex attachment and BAR proteins". The manuscript was assessed by expert reviewers, whose comments are appended to this letter. We invite you to submit a revision if you can address the reviewers' key concerns, as outlined here.

You will see that all three reviewers found this study interesting and stated that your work provides valuable new insights into cell-cell fusion mechanics and mechanisms. While reviewers #1 and #3 had only relatively minor comments about the manuscript, reviewer #2 raised several important points related to control experiments and data quantification. These should be addressed in the revised version of the manuscript by editing the manuscript text, as well as by performing the requested control experiments and clarifications related to data analysis. Reviewer #2 also stated that additional experiments employing mechanical manipulations to directly modulate membrane tension would be valuable (points #4 and #9), but these may be beyond the scope of the present study, and can also be addressed by including more discussion on these points. Finally, reviewer #2 stated that the authors should thoroughly discuss the discrepancies between their results and published work providing evidence that that ERM proteins are dispensable for macrophage migration and cellular cortex mechanics.

GENERAL GUIDELINES:

Text limits: Character count for a Report is < 20,000, not including spaces. Count includes title page, abstract, introduction, the joint Results & Discussion, and acknowledgments. Count does not include materials and methods, figure legends, references, tables, or supplemental legends.

Figures: Reports may have up to 5 main text figures. To avoid delays in production, figures must be prepared according to the policies outlined in our Instructions to Authors, under Data Presentation, <https://jcb.rupress.org/site/misc/ifora.xhtml>. All figures in accepted manuscripts will be screened prior to publication.

Supplemental information: There are strict limits on the allowable amount of supplemental data. Reports may have up to 3 supplemental figures. Up to 10 supplemental videos or flash animations are allowed. A summary of all supplemental material should appear at the end of the Materials and methods section.

Please note that JCB now requires authors to submit Source Data used to generate figures containing gels and Western blots with all revised manuscripts. This Source Data consists of fully uncropped and unprocessed images for each gel/blot displayed in the main and supplemental figures. Since your paper includes cropped gel and/or blot images, please be sure to provide one Source Data file for each figure that contains gels and/or blots along with your revised manuscript files. File names for Source Data figures should be alphanumeric without any spaces or special characters (i.e., SourceDataF#, where F# refers to the associated main figure number or SourceDataFS# for those associated with Supplementary figures). The lanes of the gels/blots should be labeled as they are in the associated figure, the place where cropping was applied should be marked (with a box), and molecular weight/size standards should be labeled wherever possible.

The typical timeframe for revisions is three to four months. If you anticipate any difficulties in meeting this aforementioned revision time limit, please contact us and we can work with you to find an appropriate time frame for resubmission. Please note that papers are generally considered through only one revision cycle, so any revised manuscript will likely be either accepted or rejected.

Thank you for this interesting contribution to Journal of Cell Biology. You can contact us at the journal office with any questions at cellbio@rockefeller.edu.

Sincerely,

Pekka Lappalainen, PhD
Monitoring Editor

Andrea L. Marat, PhD
Deputy Editor

Journal of Cell Biology

Reviewer #1 (Comments to the Authors (Required)):

This study investigates the mechanisms of cell-cell fusion in nonmuscle cells with a particular focus on the role of membrane tension controlled by actin-plasma membrane linkers. The results show that RANKL treatment of RAW 264.7 cells stimulates cell-cell fusion, downregulates ezrin expression and decreases membrane tension, whereas functional manipulations that decrease or increase the plasma membrane-to-actin linkages lead respectively to an increase or decrease of both cell-cell fusion and invadosome formation. Additionally, using a knockdown screen, the authors find that several of BAR domain-containing proteins promote cell-cell fusion and invadosome formation. The authors propose that the decreased membrane tension that results from downregulation of the membrane-actin linkages mediated by ERM proteins allows for the formation of invadosomes, likely in BAR domain-dependent manner, whereas the invadosomes in turn promote cell-cell fusion. The study is well done and provides important information about an understudied process of cell-cell fusion of nonmuscle mammalian cells. Although the part of the study dealing with the BAR domain proteins is somewhat preliminary, the current results raise important questions for subsequent research and are acceptable in the present form for this study.

I have only minor comments about the manuscript.

1. P. 5, l. 52. None of the references provided for "osteoclast and muscle fusion" report osteoclast fusion. It would be helpful to know the state of the field of osteoclast fusion, specifically.
2. P. 7, l. 89 and Fig. 1C and E. Based on the text and Fig. 1E, RANKL-treated cells that eventually fuse are larger than non-treated and not fusing cells. However, in fig. 1C, non-treated cells are the largest. Are they not representative? If so, another example should be given.
3. P. 8, ll. 96-98. The bright field microscopy (apparently, phase contrast, although no information is provided on this point) is not sufficient to make statements about filopodia or lamellipodia. F-actin labeling is required.
4. P. 8, l. 109. It would be helpful to provide brief explanation of what MA-ezrin and iMC linker are at their first appearance in the text besides the information given in the figure and Methods.
5. P. 11, ll. 154-155 and Fig. S2L. The point of knocking down Prdm1 is to see the effect on ezrin expression, so the results of RANKL+/si-Control should be statistically compared to RANKL+/si-Prdm1, but this comparison is not provided. If it is not significant, it should be said so and the results referred to as a "trend".
6. P. 11, l. 161 and Fig. 3G. Why the fusion index for si-Control in Figure 3G (mean of ~10%) is so much lower than for parental cells in Figures 2E (~65%) and for si-Control cells in figure 4A (~60%)?
7. P. 12, ll. 177-178. Please, provide common synonyms for the BAR domain proteins on the first appearance in the text.
8. P. 12, ll. 180-182. Figures S3A and B cited at the end of this sentence, give the impression that they show the TRAP activity, whereas in reality they show knockdown efficiency. Consider moving the reference to these figures to an earlier position in the sentence.
9. Fig. 5A-E. The resolution/magnification of fluorescence images is insufficient to recognize invadosomes. These images just show much actin in some cells but not others.
10. RANKL-treated si-Control cells in Figure 5C look similar to untreated parental cells in 5A (no invadosomes) but dissimilar to RANKL-treated parental cells in A, which have invadosomes. Explain.
11. It is not clear how many invadosomes a cell should have in order to be considered as invadosome-positive.
12. P. 15, l. 232. It is not obvious why fusion of cells from a macrophage-like cell line is a model for osteoclast fusion.
13. P. 15, l. 236. Is there information in the literature about expression of other BAR domain proteins in response to RANKL stimulation?

Reviewer #2 (Comments to the Authors (Required)):

The manuscript investigates the role of plasma membrane (PM) mechanics, particularly membrane-to-cortex attachment (MCA), in osteoclast fusion. The authors propose that reduced PM tension, induced by downregulation of ezrin expression following RANKL stimulation, is a prerequisite for invadosome formation and subsequent cell-cell fusion. The study uses an array of experimental approaches, including live-cell imaging, optical tweezers, and genetic interference, to explore the interplay between PM tension and the actin cytoskeleton. While the work provides interesting insights into osteoclast fusion mechanics, several points require further clarification or additional experiments.

Major Comments

1. The manuscript states that cells began to fuse approximately 60 hours after RANKL addition, but this timing likely varies between experiments. The authors should provide a standard deviation for this observation and clarify how the timing was determined. Details such as the number of fields of view analyzed, the criteria for defining "fusion," and the methodology used (e.g., visualization of membrane or cytoplasmic continuity with fluorescent markers) should be included.
2. The reported cell numbers (27-30) are low for quantifying surface tension differences between populations. The authors should provide the percentage of fused cells relative to the total population and scale their quantification accordingly. It is easy to recognize fused cells and measure tension, but comparing them to only 30 cells in the background, which might be in different cell states, is not sufficient. I am also wondering if one can really derive a total value for tension in a giant cell like the osteoclast. What would measurements from multiple locations look like? Is tension homogenous?
3. The claim that tools used to manipulate MCA do not affect differentiation (based on NFATc1 levels, Fig. S1B) is insufficiently supported by the Western blot, which is also not quantified. The authors should provide additional evidence, such as immunofluorescence data showing NFATc1 localization (nuclear vs. cytoplasmic) and examine other late-stage differentiation markers. This would strengthen their conclusion that the observed effects are independent of differentiation.
4. The lack of a direct readout for fusion, such as cytoplasmic mixing, limits the mechanistic conclusions. Transient fusion events could occur without full pore expansion, and the observed effects on membrane tension may not directly inhibit fusion. Additional experiments employing mechanical manipulations to directly modulate membrane tension would be valuable. The authors should also consider alternative explanations, such as the possibility that cortex disassembly is required for efficient delivery of fusion machinery to the membrane.
5. In Fig. 2D/E, the comparison to parental cells is not an adequate negative control. The authors should include a condition where only ezrin is overexpressed to account for potential artifacts from overexpression or transduction itself.
6. The calculation of the fusion index and its statistical presentation require clarification. Does the "n" refer to 45 fields of view per condition? The rationale for defining the index as the number of multinucleated cells per image (instead of across all images) should be provided (especially given that one fused osteoclast can fill an entire field of view, in which case the index would be 100%). The fusion indices are sometimes presented in violin plots and sometimes as a scatter plot, which can be reconciled. Additionally, the manuscript sometimes presents data from 70 hours and other times from 84 hours post-RANKL stimulation, which raises questions about whether the observed effects pertain to the ability of cells to fuse or merely the rate at which fusion occurs. For example, it is unclear whether the cells shown in Fig. 3F eventually catch up as their fusion index at 84hr would indicate.
7. It is also important to note that RANKL induces differentiation, with fusion occurring as a subsequent process. This distinction should be emphasized in the manuscript. The reported reduction in ezrin expression seems to correlate with differentiation rather than directly with fusion.
8. The authors should address discrepancies with published work <https://www.embopress.org/doi/full/10.1038/s44318-024-00173-7>. This discussion could consolidate their findings and clarify the contributions of ezrin to fusion and cell mechanics.
9. The conclusion that MCA-dependent PM tension inhibits invadosome formation and fusion is overstated without direct evidence. The authors should include additional experiments demonstrating the inhibition of invadosomes under conditions of elevated PM tension to strengthen this claim (see comment 4). The fact that there is a measurable effect on membrane tension does not immediately lead to the conclusion that membrane tension significantly opposes invadopodia formation without additional experiments.

Minor Comments

1. Figure 1B and Video 1: The pseudo-coloring in Fig. 1B and Video 1 is unhelpful and obscures the images. Consider using dashed lines or arrowheads to mark cells. Additionally, clarify the time labels; for example, in panel B, "time 0" is ambiguous, as it represents fusion in the bottom panel but not in others.
2. Lines 97-98: The phrase "for efficient cell-cell fusion" should be omitted, as this is an unsupported assumption.
3. Please provide a reference for the statement that DC-STAMP is a master regulator of cell-cell fusion. As far as I am aware, it is a key regulator of differentiation.
4. Lines 195-196: There is no measurement provided to support the claim of "exponential progression" of fusion at 84 hours.

Reviewer #3 (Comments to the Authors (Required)):

The paper by Wan and colleagues interrogates the interplay between membrane tension, MAC (Membrane-Actin Contacts) and

BAR domain proteins in the osteoclast cell-cell fusion induced by RANKL. The paper is concise, well-driven and with clear data. The Authors show:

-RANKL treatment induces a drop in membrane tension required for fusion.

-This drop is caused by the degradation of Ezrin, that links membrane to actin cortex. SiRNA of ezrin is sufficient to induce membrane tension drop and cell fusion.

-The drop (and subsequent fusion) allows for invadosome formation by BAR proteins and actin polymerization. Knock-outs of essential proteins for invadosome formation reduces cell-cell fusion.

As such the paper is a well conducted study and could be published as it is. I suggest only 1 experiment (not required for publication). The authors propose that membrane cortex attachment and invadosome formation are coupled through membrane tension. To further test their proposition, I would suggest to:

1-to monitor tension in si-Fnbp1 cells treated with RANKL (or other si with similar cell-cell fusion scores) cells. 2 effects could be seen: a lower membrane tension than in normal cells, suggesting that invadosomes partially restore the tension drop generated by MCA detachment. Secondly, the author could see a higher tension than in control cells, supporting the view that if invadosomes are not formed, MCA can reform quickly and increase tension again. Those 2 observations would support the claim of the authors.

We would like to express our sincere gratitude to the reviewers who took the time to evaluate our paper. All the comments are very valuable, and responding to them has allowed us to greatly improve the manuscript. Below, we have responded to each of the comments.

Reviewer #1 (Comments to the Authors (Required)):

This study investigates the mechanisms of cell-cell fusion in nonmuscle cells with a particular focus on the role of membrane tension controlled by actin-plasma membrane linkers. The results show that RANKL treatment of RAW 264.7 cells stimulates cell-cell fusion, downregulates ezrin expression and decreases membrane tension, whereas functional manipulations that decrease or increase the plasma membrane-to-actin linkages lead respectively to an increase or decrease of both cell-cell fusion and invadosome formation. Additionally, using a knockdown screen, the authors find that several of BAR domain-containing proteins promote cell-cell fusion and invadosome formation. The authors propose that the decreased membrane tension that results from downregulation of the membrane-actin linkages mediated by ERM proteins allows for the formation of invadosomes, likely in BAR domain-dependent manner, whereas the invadosomes in turn promote cell-cell fusion. The study is well done and provides important information about an understudied process of cell-cell fusion of nonmuscle mammalian cells. Although the part of the study dealing with the BAR domain proteins is somewhat preliminary, the current results raise important questions for subsequent research and are acceptable in the present form for this study.

I have only minor comments about the manuscript.

- 1. P. 5, l. 52. None of the references provided for "osteoclast and muscle fusion" report osteoclast fusion. It would be helpful to know the state of the field of osteoclast fusion, specifically.*

We appreciate the input. As requested, we have included three references related to osteoclast fusion (Oikawa 2012, Sør 2015, Wang 2015) in the revised manuscript (lines 53-54).

- 2. P. 7, l. 89 and Fig. 1C and E. Based on the text and Fig. 1E, RANKL-treated cells that eventually fuse are larger than non-treated and not fusing cells. However, in fig. 1C, non-treated cells are the largest. Are they not representative? If so, another example should be given.*

As the reviewer correctly noted, the images of non-treated cells in Fig. 1B were not representative. We apologize for the confusion and have replaced them with more representative images in the revised manuscript.

3. *P. 8, ll. 96-98. The bright field microscopy (apparently, phase contrast, although no information is provided on this point) is not sufficient to make statements about filopodia or lamellipodia. F-actin labeling is required.*

We have performed F-actin staining, and the results are shown in the new Fig. S1C. While the cells before RANKL treatment showed F-actin-enriched filopodia, the cells after RANKL treatment showed invadosomes at the extended, flattened membrane structure. We would like to thank the reviewer for this valuable input, as it has enabled us to discern that these structures are distinct from conventional lamellipodia. Instead, these structures appear to reflect the observations consistent with our data presented in Fig. 5. In light of these findings, the term "lamellipodia" has been removed from the entire text of the revised manuscript. Furthermore, the figure legend for Fig. 1B has been updated to specify that the images were captured using a phase contrast microscope (line 698).

4. *P. 8, l. 109. It would be helpful to provide brief explanation of what MA-ezrin and iMC linker are at their first appearance in the text besides the information given in the figure and Methods.*

As suggested, a more detailed explanation about the two molecular tools is added to the main text of the revised manuscript (lines 118-122).

5. *P. 11, ll. 154-155 and Fig. S2L. The point of knocking down Prdm1 is to see the effect on ezrin expression, so the results of RANKL+/si-Control should be statistically compared to RANKL+/si-Prdm1, but this comparison is not provided. If it is not significant, it should be said so and the results referred to as a "trend".*

Thank you for the advice. We have statistically compared between the ezrin expression in RANKL+/si-Control and RANKL+/si-Prdm1 conditions, and found there was no significant difference. The result is shown as "ns" in Fig. S3L, and the text is corrected accordingly (lines 169-171).

6. *P. 11, l. 161 and Fig. 3G. Why the fusion index for si-Control in Figure 3G (mean of ~10%) is so much lower than for parental cells in Figures 2E (~65%) and for si-Control cells in figure 4A (~60%)?*

We apologize for the insufficient description of the experimental procedures concerning these

results. In order to examine whether the ezrin depletion facilitates cell-cell fusion, the samples were fixed at an earlier time point (70 h) than those shown in Fig. 2 and 4 (84 h). Fig. 3G demonstrates the knockdown cells show higher fusion rates than control cells as early as 70 h after RANKL treatment, supporting our hypothesis. We have corrected the text (lines 175-178) and added the time point ("70 h") in Fig. 3G.

7. P. 12, ll. 177-178. Please, provide common synonyms for the BAR domain proteins on the first appearance in the text.

As suggested, we have added common synonyms in parallel with gene names of the BAR domain proteins in the revised manuscript (line 194).

8. P. 12, ll. 180-182. Figures S3A and B cited at the end of this sentence, give the impression that they show the TRAP activity, whereas in reality they show knockdown efficiency. Consider moving the reference to these figures to an earlier position in the sentence.

We appreciate the advice. The part citing Fig. S3A and B (Fig. S3 N-P in the revised manuscript) is moved to an earlier position (line 197).

9. Fig. 5A-E. The resolution/magnification of fluorescence images is insufficient to recognize invadosomes. These images just show much actin in some cells but not others.

We agree with the reviewer and therefore present higher resolution images in the revised manuscript (Fig. 5A, C, and E). The differences between the previous and revised versions of the images are as follows.

	Microscope	Objective lens	Image size	µm/pixel	Visualization
Original submission	Andor BC43	40x/0.95	2040*1992	0.1559	Single slice
Revised manuscript	Andor BC43	60x/1.42	2040*2040	0.1040	Max intensity projection

10. RANKL-treated si-Control cells in Figure 5C look similar to untreated parental cells in 5A (no invadosomes) but dissimilar to RANKL-treated parental cells in A, which have invadosomes. Explain.

We apologize for the confusion. As stated in the figure legends, these two data were obtained at different time points (72 h after RANKL treatment for Fig. 5A, while 60 h for Fig. 5C). To clarify this, the time points of RANKL treatment are labeled at the bottom of each graph in the revised manuscript. In addition, the image of si-Control cells in Fig. 5C did not represent the typical appearance of this condition and has been replaced with a more appropriate image.

11. It is not clear how many invadosomes a cell should have in order to be considered as invadosome-positive.

Again, we apologize for the lack of clarity on this point. Invadosomes were defined as the accumulation of more than 20 actin dots per cell or an actin cluster with a diameter of more than 5 μm including the actin ring. In the revised manuscript, we add a detailed explanation of invadosome quantification in the Materials and methods section (lines 482-486).

12. P. 15, l. 232. It is not obvious why fusion of cells from a macrophage-like cell line is a model for osteoclast fusion.

Thank you for the input. We have added a brief explanation of osteoclast differentiation from the macrophage lineage in the Introduction (line 36).

13. P. 15, L. 236. Is there information in the literature about expression of other BAR domain proteins in response to RANKL stimulation?

In the cited literature (Oikawa and Matsuo, 2012), other BAR domain proteins such as srGAP3 and Fnbp1 also showed changes in their transcript levels. While srGAP3 showed a gradual increase in its expression level, the transcript of Fnbp1 is lost at 24 h after RANKL treatment and returns to the initial state at 48 h. The fact that si-Srgap3 tended to increase the fusion index (Fig. 4A) suggests that there may be a coordinated regulation of the expression levels of multiple BAR domain proteins during osteoclast fusion. We thank the reviewer for this valuable question.

Reviewer #2 (Comments to the Authors (Required)):

The manuscript investigates the role of plasma membrane (PM) mechanics, particularly membrane-to-cortex attachment (MCA), in osteoclast fusion. The authors propose that reduced PM tension, induced by downregulation of ezrin expression following RANKL stimulation, is a prerequisite for

invadosome formation and subsequent cell-cell fusion. The study uses an array of experimental approaches, including live-cell imaging, optical tweezers, and genetic interference, to explore the interplay between PM tension and the actin cytoskeleton. While the work provides interesting insights into osteoclast fusion mechanics, several points require further clarification or additional experiments.

Major Comments

- 1. The manuscript states that cells began to fuse approximately 60 hours after RANKL addition, but this timing likely varies between experiments. The authors should provide a standard deviation for this observation and clarify how the timing was determined. Details such as the number of fields of view analyzed, the criteria for defining "fusion," and the methodology used (e.g., visualization of membrane or cytoplasmic continuity with fluorescent markers) should be included.*

As the reviewers correctly point out, there is some variation in the timing of cell fusion from experiment to experiment. Therefore, we examined the timing of cell fusion in detail (Fig. S1A). Fig. S1B shows the median of the fusion index at each time point after RANKL treatment along with the standard deviation. It was found that cell fusion begins at 48 hours, earlier than 60 hours after RANKL treatment, and the text has been revised accordingly (lines 83-85). We thank the reviewer for this valuable input. We have added a more detailed explanation of the method for determining the cell fusion index, including the method for recognizing cell boundaries, the area of the region to be quantified, and the number of data points, to the Materials and methods section (lines 460-462, 468-470).

- 2. The reported cell numbers (27-30) are low for quantifying surface tension differences between populations. The authors should provide the percentage of fused cells relative to the total population and scale their quantification accordingly. It is easy to recognize fused cells and measure tension, but comparing them to only 30 cells in the background, which might be in different cell states, is not sufficient. I am also wondering if one can really derive a total value for tension in a giant cell like the osteoclast. What would measurements from multiple locations look like? Is tension homogenous?*

We apologize for the confusion. The data presented only measured mononuclear cells (not multinuclear cells) after RANKL treatment, because we found that RANKL-treated cells exhibited a protrusive phenotype prior to cell-cell fusion, and therefore considered a decrease in MCA to be a prerequisite for cell fusion (Fig. 1B). To avoid this confusion, we have modified the text to indicate that we measured the tether force of mononuclear cells (lines 110-111). It is difficult to confirm whether individual cells actually responded to RANKL treatment. Therefore, we considered that the collection of measurements should reflect the physical properties of RAW

cells after RANKL treatment as a whole. We believe that the number of cells measured for tether force is comparable to other papers in the field (JCB 2001 PMID: 10629223, PNAS 2013 PMID: 28687667, Cell Stem Cell 2021 PMID: 33207217), and is an appropriate sample size for the collection of measurements. We measured PM tension on the lateral side of cells, because extended membranes (as observed in RANKL-treated cells) compromise force measurement by optical trapping. Since PM tension is known to correlate with pERM levels, and RANKL (-) cells exhibit overall higher pERM levels throughout the plasma membrane (Fig. 3D), we believe that optical tweezers measurements reflect the overall PM tension at least in unstimulated cells. In addition, PM tension has been reported to be instantaneously and globally propagated throughout the cell (Cell 2023, PMID: 37311454), suggesting the homogeneity of the measurement.

3. *The claim that tools used to manipulate MCA do not affect differentiation (based on NFATc1 levels, Fig. S1B) is insufficiently supported by the Western blot, which is also not quantified. The authors should provide additional evidence, such as immunofluorescence data showing NFATc1 localization (nuclear vs. cytoplasmic) and examine other late-stage differentiation markers. This would strengthen their conclusion that the observed effects are independent of differentiation.*

In response to this criticism, we present three additional data in the revised manuscript. First, Western blotting data of NFATc1 expression were quantified and statistically tested (Fig. S2C). Second, the intracellular localization of NFATc1 was evaluated by confocal microscopy (Fig. S2D) and the ratio of nuclear to cytoplasmic signal intensity was quantified (Fig. S2E). It was shown that cell lines expressing ezrin, MA-ezrin, and iMC showed similar or even higher levels of nuclear translocation of NFATc1 compared to parental cells. Interestingly, nuclear NFATc1 is evident in ezrin, MA-ezrin, and iMC-expressing cells even in the absence of RANKL treatment, although the details of this apparent feedback mechanism are beyond the scope of the current study. Third, we added data on TRAP staining of each cell line as a late-stage differentiation marker (Fig. S2F). These results collectively demonstrate that the suppression of cell-cell fusion by forced elevation of MCA is not due to suppression of differentiation.

4. *The lack of a direct readout for fusion, such as cytoplasmic mixing, limits the mechanistic conclusions. Transient fusion events could occur without full pore expansion, and the observed effects on membrane tension may not directly inhibit fusion. Additional experiments employing mechanical manipulations to directly modulate membrane tension would be valuable. The authors should also consider alternative explanations, such as the possibility that cortex disassembly is required for efficient delivery of fusion machinery to the membrane.*

We thank the reviewer for these suggestions. Currently, the gold standard for increasing MCA-mediated tension is the direct manipulation of MCA by the molecular tools used in this study, and is widely accepted in the field. Other methods to constantly increase PM tension, such as cholesterol depletion and expression of active RhoA, are considered, but these manipulations are likely to affect unintended signaling pathways other than tension regulation. We hope that other tools will be developed in the future that can directly manipulate MCA. Additionally, we have already mentioned the possible relationship between cortex disassembly and fusion events in the discussion part, citing the relevant paper (lines 274-277).

5. *In Fig. 2D/E, the comparison to parental cells is not an adequate negative control. The authors should include a condition where only ezrin is overexpressed to account for potential artifacts from overexpression or transduction itself.*

We agree with the reviewer's comments and have actually compared the fusion index of parental cells and ezrin-expressing cells. As shown below, there is no statistical difference between them, so we can conclude that there is no effect of overexpression or gene transduction. However, this experimental data was not obtained at the same time as the experiments shown in Fig. 2D and E, so it cannot be combined with the main figure. We are willing to provide these data as supplementary material if requested by the reviewers and editors.

6. *The calculation of the fusion index and its statistical presentation require clarification. Does the "n" refer to 45 fields of view per condition? The rationale for defining the index as the number of multinucleated cells per image (instead of across all images) should be provided (especially given that one fused osteoclast can fill an entire field of view, in which case the index would be 100%). The fusion indices are sometimes presented in violin plots and sometimes as a scatter plot, which can be reconciled. Additionally, the manuscript sometimes presents data from 70 hours and other times from 84 hours post-RANKL stimulation, which raises questions about whether the observed effects pertain to the ability of cells to fuse or merely the rate at which fusion occurs. For example, it is unclear whether the cells shown in Fig. 3F eventually catch up as their fusion index at 84hr would indicate.*

We apologize for the insufficient description of the fusion index and its presentation. As the reviewer correctly noted, the "n" corresponds to the number of fields of view. In this study, we applied the quantification method described in the paper by Shilagardi et al. (Science 340, 6130 (2013): 359–63. PMID: 23470732), which is now cited in the Materials and methods section of the revised manuscript (line 469). According to the advice, we have reconciled the fusion index data to be presented only in violin plots. Regarding the reviewer's concern about whether knockdowns affect the ability of cells to fuse or the rate of fusion, our data show that the fusion index of si-Control cells eventually catches up to the si-Ezrin or si-ERM cells at 84 h after RANKL treatment, supporting the latter possibility. We are willing to provide these data as supplementary material if requested by the reviewers and editors.

7. *It is also important to note that RANKL induces differentiation, with fusion occurring as a*

subsequent process. This distinction should be emphasized in the manuscript. The reported reduction in ezrin expression seems to correlate with differentiation rather than directly with fusion.

We found this point very important, therefore avoid using the term "RANKL stimulation," and instead use the term "RANKL treatment" or "RANKL-induced osteoclast differentiation" in the revised manuscript. Thank you for this valuable suggestion.

8. *The authors should address discrepancies with published work <https://www.embopress.org/doi/full/10.1038/s44318-024-00173-7>. This discussion could consolidate their findings and clarify the contributions of ezrin to fusion and cell mechanics.*

Thank you for pointing this out. In this paper, it was reported that pERM level is decreased during M-CSF-stimulated differentiation of mouse progenitor cells (round) into macrophages (spreading), consistent with our study and other literature showing that decreased pERM level is correlated with cell spreading. In spreading macrophages, further loss of ERM proteins might have little effect on their motility and membrane mechanics. We found that in the differentiation of RAW cells into osteoclasts, only ezrin is decreased, consistent with the data found in this paper in which ezrin is also decreased in differentiation to spreading macrophages. However, in this paper, the expression of moesin seems to be increased instead. We have observed that a reduction in ezrin alone promotes cell fusion (Fig. 3F and G), suggesting that ezrin may play a major role in the regulation of MCA, at least in fusion-competent macrophages. We have added these points in the discussion part (lines 239-242).

9. *The conclusion that MCA-dependent PM tension inhibits invadosome formation and fusion is overstated without direct evidence. The authors should include additional experiments demonstrating the inhibition of invadosomes under conditions of elevated PM tension to strengthen this claim (see comment 4). The fact that there is a measurable effect on membrane tension does not immediately lead to the conclusion that membrane tension significantly opposes invadopodia formation without additional experiments.*

As the reviewer pointed out, our conclusion would not be fully supported until a method other than the molecular tools used in this study that allows direct manipulation of PM tension becomes available (please see our response to comment #4). We state the limitation of this study based on this discussion in the revised manuscript (lines 233-235).

Minor Comments

1. *Figure 1B and Video 1: The pseudo-coloring in Fig. 1B and Video 1 is unhelpful and obscures the images. Consider using dashed lines or arrowheads to mark cells. Additionally, clarify the time labels; for example, in panel B, "time 0" is ambiguous, as it represents fusion in the bottom panel but not in others.*

In response, we have improved the clarity of the time-lapse images in Fig. 1B. Specifically, instead of coloring the entire cell, we have indicated the cell with a colored arrowhead, making it easier to see the shape and outline of the cell. We also changed the display to use frame numbers instead of time to avoid confusion. On the other hand, please note that in Fig. 1E and F, the moment of fusion is shown as "0 h" in order to compare the fused cell with other conditions at the same time. We thank the reviewer for this helpful comment.

2. *Lines 97-98: The phrase "for efficient cell-cell fusion" should be omitted, as this is an unsupported assumption.*

We have corrected this phrase to "before cell-cell fusion" (line 105). Thank you for the suggestion.

3. *Please provide a reference for the statement that DC-STAMP is a master regulator of cell-cell fusion. As far as I am aware, it is a key regulator of differentiation.*

As the reviewer correctly commented, DC-STAMP should be referred to as "a key regulator". The corresponding part is corrected (lines 195-196).

4. *Lines 195-196: There is no measurement provided to support the claim of "exponential progression" of fusion at 84 hours.*

We have deleted this sentence in response to this criticism (line 211).

Reviewer #3 (Comments to the Authors (Required)):

The paper by Wan and colleagues interrogates the interplay between membrane tension, MAC (Membrane-Actin Contacts) and BAR domain proteins in the osteoclast cell-cell fusion induced by

RANKL. The paper is concise, well-driven and with clear data. The Authors show:

-RANKL treatment induces a drop in membrane tension required for fusion.

-This drop is caused by the degradation of Ezrin, that links membrane to actin cortex. SiRNA of ezrin is sufficient to induce membrane tension drop and cell fusion.

-The drop (and subsequent fusion) allows for invadosome formation by BAR proteins and actin polymerization. Knock-outs of essential proteins for invadosome formation reduces cell-cell fusion.

As such the paper is a well conducted study and could be published as it is. I suggest only 1 experiment (not required for publication). The authors propose that membrane cortex attachment and invadosome formation are coupled through membrane tension. To further test their proposition, I would suggest to:

1-to monitor tension in si-Fnbp1 cells treated with RANKL (or other si with similar cell-cell fusion scores) cells. 2 effects could be seen: a lower membrane tension than in normal cells, suggesting that invadosomes partially restore the tension drop generated by MCA detachment. Secondly, the author could see a higher tension than in control cells, supporting the view that if invadosomes are not formed, MCA can reform quickly and increase tension again. Those 2 observations would support the claim of the authors.

We are grateful to the reviewer for his/her positive evaluation of our paper, and we appreciate the insight into the potential for a mutual regulation between invadosome formation and PM tension. Since BAR domain proteins such as Fnbp1/FBP17 are factors downstream of PM tension fluctuations, we had not previously considered the possibility of such a feedback loop. Moving forward, we are committed to pursuing research from this perspective, and we deeply appreciate the valuable advice that has guided our further exploration.

March 31, 2025

RE: JCB Manuscript #202411024R

Toshiki Itoh
Kobe University

Dear Prof. Itoh:

Thank you for submitting your revised manuscript entitled "Mechanical control of osteoclast fusion by membrane-cortex attachment and BAR proteins". We would be happy to publish your paper in JCB pending final revisions necessary to meet our formatting guidelines (see details below).

A. MANUSCRIPT ORGANIZATION AND FORMATTING:

- 1) Text limits: Character count for Reports is < 20,000, not including spaces. Count includes abstract, introduction, combined results and discussion, and acknowledgments. Count does not include title page, figure legends, materials and methods, references, tables, or supplemental legends.
- 2) Figures limits: Articles may have up to 10 main text figures.
- 3) Figure formatting: Scale bars must be present on all microscopy images, including inset magnifications. Molecular weight or nucleic acid size markers must be included on all gel electrophoresis. Aspect ratios of images may not be altered.
- 4) Statistical analysis: Error bars on graphic representations of numerical data must be clearly described in the figure legend. The number of independent data points (n) represented in a graph must be indicated in the legend. Statistical methods should be explained in full in the materials and methods. For figures presenting pooled data the statistical measure should be defined in the figure legends. Please also be sure to indicate the statistical tests used in each of your experiments (either in the figure legend itself or in a separate methods section) as well as the parameters of the test (for example, if you ran a t-test, please indicate if it was one- or two-sided, etc.). Also, if you used parametric tests, please indicate if the data distribution was tested for normality (and if so, how). If not, you must state something to the effect that "Data distribution was assumed to be normal but this was not formally tested."
- 5) Abstract and title: The abstract should be no longer than 160 words and should communicate the significance of the paper for a general audience. The title should be less than 100 characters including spaces. Make the title concise but accessible to a general readership.
- 6) Materials and methods: Should be comprehensive and not simply reference a previous publication for details on how an experiment was performed. Please provide full descriptions in the text for readers who may not have access to referenced manuscripts.
- 7) All antibodies, cell lines, animals, and tools used in the manuscript should be described in full, including accession numbers for materials available in a public repository such as the Resource Identification Portal. Please be sure to provide the sequences for all of your primers/oligos and RNAi constructs in the materials and methods. You must also indicate in the methods the source, species, and catalog numbers (where appropriate) for all of your antibodies. Please also indicate the acquisition and quantification methods for immunoblotting/western blots.
- 8) Microscope image acquisition: The following information must be provided about the acquisition and processing of images:
 - a. Make and model of microscope
 - b. Type, magnification, and numerical aperture of the objective lenses
 - c. Temperature
 - d. Imaging medium
 - e. Fluorochromes
 - f. Camera make and model
 - g. Acquisition software
 - h. Any software used for image processing subsequent to data acquisition. Please include details and types of operations involved (e.g., type of deconvolution, 3D reconstitutions, surface or volume rendering, gamma adjustments, etc.).

10) Supplemental materials: There are strict limits on the allowable amount of supplemental data. Reports may have up to 3 supplemental figures. Please also note that tables, like figures, should be provided as individual, editable files. A summary of all supplemental material should appear at the end of the Materials and methods section.

13) ORCID IDs: ORCID IDs are unique identifiers allowing researchers to create a record of their various scholarly contributions in a single place. Please note that ORCID IDs are now *required* for all authors. At resubmission of your final files, please be sure to provide your ORCID ID and those of all co-authors.

Please note that JCB now requires authors to submit Source Data used to generate figures containing gels and Western blots with all revised manuscripts. This Source Data consists of fully uncropped and unprocessed images for each gel/blot displayed in the main and supplemental figures. For assays performed using capillary electrophoresis and/or immunoassay-based detection, authors should instead provide the electropherogram graph(s) for each experiment, plotting fluorescence/chemiluminescence intensity vs. molecular weight/size. Please be sure to provide one Source Data file for each figure gels, blots, and/or capillary electrophoresis assays along with your revised manuscript files. File names for Source Data figures should be alphanumeric without any spaces or special characters (i.e., SourceDataF#, where F# refers to the associated main figure number or SourceDataFS# for those associated with Supplementary figures). For traditional gels and blots, the lanes of the gels/blots should be labeled as they are in the associated figure, the place where cropping was applied should be marked (with a box), and molecular weight/size standards should be labeled wherever possible. For capillary electrophoresis assays, each trace in the graph should be color-coded and labeled to indicate which protein, gene, or sample is being measured (please try to avoid red/green combinations to accommodate our color-blind readers).

Journal of Cell Biology now requires a data availability statement for all research article submissions. These statements will be published in the article directly above the Acknowledgments. The statement should address all data underlying the research presented in the manuscript. Please visit the JCB instructions for authors for guidelines and examples of statements at (<https://rupress.org/jcb/pages/editorial-policies#data-availability-statement>).

B. FINAL FILES:

**It is JCB policy that if requested, original data images must be made available to the editors. Failure to provide original images

upon request will result in unavoidable delays in publication. Please ensure that you have access to all original data images prior to final submission.**

Thank you for your attention to these final processing requirements. Please revise and format the manuscript and upload materials within 7 days. If you need an extension for whatever reason, please let us know and we can work with you to determine a suitable revision period.

Thank you for this interesting contribution, we look forward to publishing your paper in Journal of Cell Biology.

Sincerely,

Pekka Lappalainen, PhD
Monitoring Editor

Andrea L. Marat, PhD
Deputy Editor

Journal of Cell Biology

Reviewer #2 (Comments to the Authors (Required)):

The authors have addressed all my comments and I do not have any further suggestions.